# Maternal aryl hydrocarbon receptor activation protects newborns against necrotizing enterocolitis

Peng Lu[1,3]✉, Yukihiro Yamaguchi[1], William B. Fulton[1], Sanxia Wang[1], Qinjie Zhou[1], Hongpeng Jia[1], Mark L. Kovler[1], Andres Gonzalez Salazar [1], Maame Sampah [1], Thomas Prindle Jr.[1], Peter Wipf[2], Chhinder P. Sodhi[1,3] & David J. Hackam [1]✉

Necrotizing enterocolitis (NEC) is a disease of premature infants characterized by acute intestinal necrosis. Current dogma suggests that NEC develops in response to post-natal dietary and bacterial factors, and so a potential role for in utero factors in NEC remains unexplored. We now show that during pregnancy, administration of a diet rich in the aryl hydrocarbon receptor (AHR) ligand indole-3-carbinole (I3C), or of breast milk, activates AHR and prevents NEC in newborn mice by reducing Toll-like receptor 4 (TLR4) signaling in the newborn gut. Protection from NEC requires activation of AHR in the intestinal epithelium which is reduced in mouse and human NEC, and is independent of leukocyte activation. Finally, we identify an AHR ligand ("A18") that limits TLR4 signaling in mouse and human intestine, and prevents NEC in mice when administered during pregnancy. In summary, AHR signaling is critical in NEC development, and maternally-delivered, AHR-based therapies may alleviate NEC.

[1] Division of Pediatric Surgery, Johns Hopkins University School of Medicine and the Johns Hopkins Children's Center, Baltimore, MD, USA. [2] Department of Chemistry, University of Pittsburgh, Pittsburgh, PA, USA. [3] These authors contributed equally: Peng Lu and Chhinder P. Sodhi. ✉email: peng.lu@jhmi.edu; Dhackam1@jhmi.edu

Necrotizing enterocolitis (NEC) is an often fatal disease of premature infants that is characterized by the acute onset of inflammation and necrosis of the intestine, leading to overwhelming sepsis and death[1,2]. NEC develops in the setting of premature birth, the administration of formula feeds, and bacterial colonization of the newborn gastrointestinal tract, and caries a 30% mortality rate[3]. In seeking to understand the pathogenesis of NEC, we[4–6] and others[7,8] have shown that exaggerated signaling of the receptor for bacterial lipopolysaccharide, namely toll-like receptor 4 (TLR4), on the intestinal epithelium[5] plays a critical role in NEC development. TLR4 is expressed at higher levels in the premature as compared with the full-term intestinal epithelium[5,9], and its activation by luminal bacteria leads to mucosal death and bacterial translocation[10,11]. Current dogma suggests that NEC develops in response to dietary and bacteriologic factors that are present in the postnatal period, which explains why all preventive strategies have so far been targeted after birth[12,13]. However, there is emerging evidence to question this dogma, and to raise the possibility that NEC may also reflect an in utero process, and by extension, to suggest that interventions in the prenatal period could prevent NEC development. For example, NEC occurs more frequently and with greater severity in babies who are born after in utero bacterial infection[14,15], while certain maternal diets have been shown to reduce the incidence of premature birth and NEC[1,16,17].

Based upon these observations, we now hypothesize that maternal–fetal signaling can modulate the pathogenesis of NEC, and that a window of opportunity may exist in utero to prevent this disease. To test this hypothesis, we focus here on the aryl hydrocarbon receptor (AHR), a ligand-activated transcription factor that recognizes environmental and dietary ligands[18], including those present in green leafy vegetables[19,20], and which has been shown to induce immune protection[21].

In this work, we now reveal that the administration of a maternal diet that is rich in the AHR ligand indole-3-carbinol (I3C) during pregnancy can prevent NEC in the offspring, and that subsequent AHR signaling in the newborn intestinal epithelium prevents NEC by curtailing the extent of TLR4 signaling. We also show that breast milk prevents NEC through activation of AHR in the newborn gut and thus reducing TLR4 signaling in the newborn intestinal epithelium. Finally, using a screen of clinically relevant compounds, we identify an AHR ligand ("A18") that can activate AHR and limit TLR4 in human tissue, thus serving as a potential NEC prevention agent when administered in utero. Taken together, these findings establish a critical link between maternal–fetal AHR signaling and NEC prevention, and highlight a role for AHR in the pathogenesis and treatment of this devastating disease.

## Results

**Maternal administration of the AHR ligand I3C during pregnancy prevents NEC in mice.** We first sought to investigate whether activation of AHR during pregnancy could prevent NEC in newborn mice, and therefore used the experimental design in Fig. 1a. Supplementation of the maternal diet with the AHR ligand I3C throughout pregnancy induced the expression of the AHR response gene *Cyp1a1* in the small intestines of both the mother (Fig. 1b) and the fetus at e17.5 (Fig. 1c), confirming that AHR activation occurs in both the mother and fetus after in utero treatment. Continuing the administration of I3C to the lactating mother after delivery was also found to induce the expression of *Cyp1a1* in the intestines of suckling pups at p1 and p11 (Fig. 1d, e). Next, to determine whether maternal AHR activation could prevent NEC, we administered either I3C or a control diet that lacked I3C to mice during both pregnancy and lactation, and then

induced NEC in the pups using a well-validated model[22,23] that closely mimics the human disease (see Methods). As shown in Fig. 1f–h, the oral administration of I3C during pregnancy significantly reduced the severity of experimental NEC in the pups, as manifest by reduced histological injury (Fig. 1f), reduced blinded NEC severity score (Fig. 1g), and reduced expression of *Il6* (Fig. 1h) and *Tnf-α* (Fig. 1i) in the intestinal epithelium.

In order to determine whether maternal-derived AHR ligands were present in tissues to which the developing fetus would be exposed, we harvested the amniotic fluid, breast milk, and serum from pregnant mice that had been administered either I3C-enriched or a control diet. These maternal-derived fluids were then incubated with IEC-6 enterocytes and assessed for AHR activation. As shown in Fig. 1j–l, maternal-derived fluids from mothers that were fed an I3C-enriched diet but not a control diet resulted in a significant increase in the expression of *Cyp1a1* in IEC-6 cells, indicating that these fluids contain maternal-derived AHR ligands.

Given the critical importance of TLR4 signaling to NEC pathogenesis[4,5,11,24], we next sought to determine whether the maternal administration of I3C could blunt TLR4 signaling in the pup intestine. As shown in Fig. 1m, n, maternal administration of I3C significantly reduced the induction of the pro-inflammatory cytokines *Il6* and *Tnf-α* in the neonatal (p11) mouse intestine in response to the administration of the specific TLR4 ligand lipopolysaccharides (LPS), as compared to pups from non I3C-fed mothers, supporting a protective effect of maternal AHR activation on the TLR4 responsiveness of the neonatal pup.

Taken together, these findings reveal that maternal delivery of AHR ligands to the fetus can prevent the development of NEC. We next examined the site of AHR signaling in the newborn mice, and the mechanisms mediating its protective effects on the development of NEC.

**AHR is expressed on newborn intestinal epithelium where its activation protects against NEC.** To examine how the administration of AHR ligands during pregnancy can prevent NEC, we next explored the expression of AHR in the small intestines of humans, mice, and piglets with and without NEC. As shown in Fig. 2, the expression of AHR was significantly reduced in the small intestine of infants with NEC as compared with control infants (Fig. 2a–c), in the intestinal mucosa of mice that were induced to develop NEC (Fig. 2d–f), and in an experimental model of NEC in premature piglets (Fig. 2g–i), compared to control subjects. These findings suggest a universal response of AHR reduction in NEC that is not specific to one species. To test whether a lack of AHR signaling was a cause and not a consequence of NEC, we induced NEC in mice from which we had deleted *Ahr* globally (*Ahr*[−/−]), or specifically from the intestinal epithelium (*Ahr*[ΔIEC]) or the myeloid compartment (*Ahr*[Δlys]). Confirmation of cell-specific *Ahr* deletion in each strain is shown in Supplementary Fig. 1. As shown in Fig. 2, *Ahr*[−/−] mice developed significantly more severe NEC than wild-type mice, as determined by higher histologic grade (Fig. 2j, k) and higher expression of the pro-inflammatory cytokine genes *Il6* and *Tnf-α* in the intestine (Fig. 2l, m). Importantly, AHR signaling on the intestinal epithelium as opposed to leukocytes was required for the protection from NEC, as *Ahr*[ΔIEC] mice had more severe NEC compared to wild-type mice (Fig. 2j–m), while NEC severity was not different between wild-type mice and *Ahr*[Δlys] mice which lack AHR on leukocytes, but which express AHR on the intestinal epithelium (as seen in Supplementary Fig. 1).

Based on the above findings, and to investigate whether activating AHR on the newborn intestinal epithelium could prevent NEC, we next explored whether feeding neonatal mice an AHR ligand could reduce NEC severity. To do so, we developed

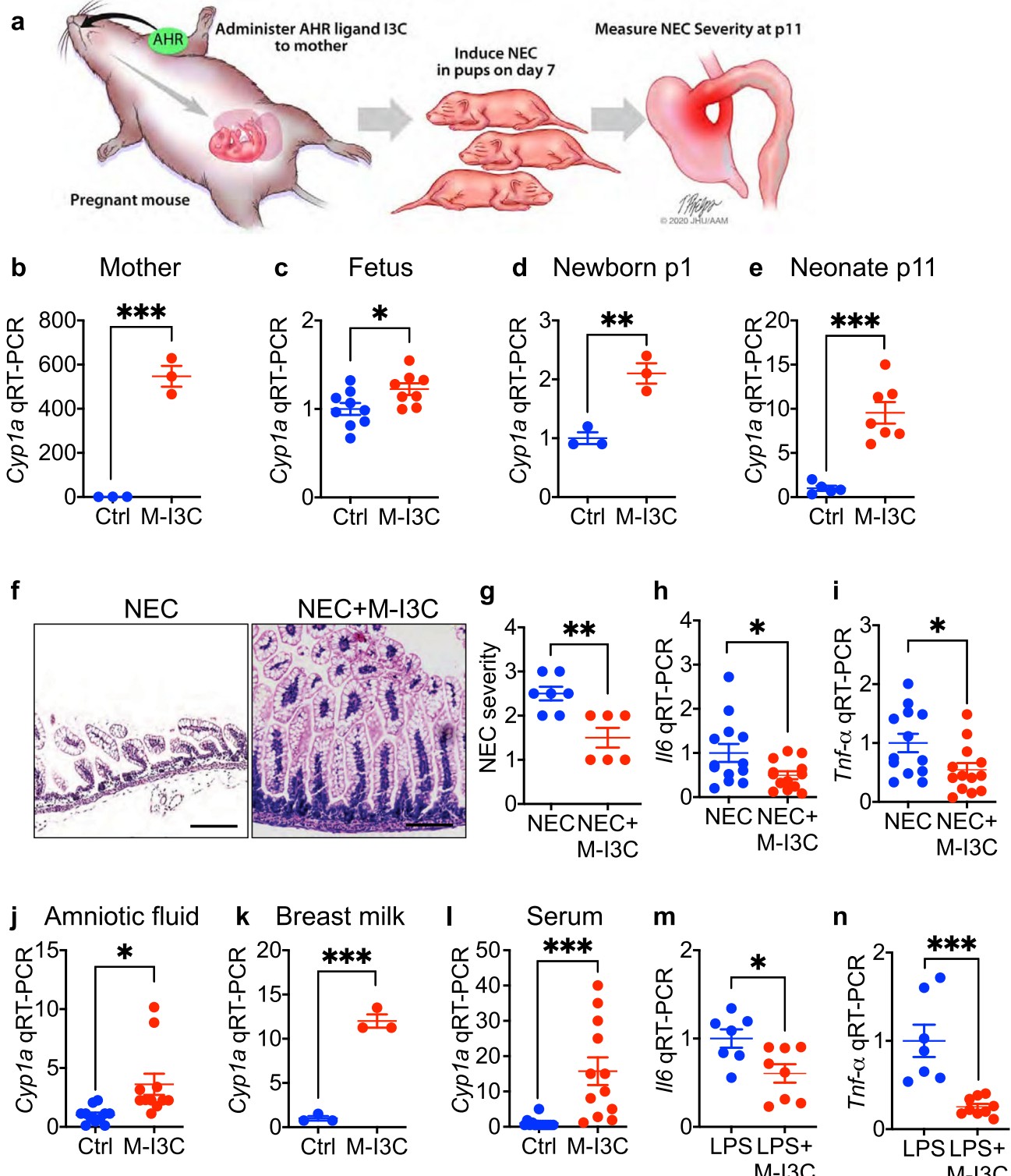

an AHR-ligand rich infant diet by supplementing the infant formula with I3C[19], which was then administered to pups in our mouse NEC model (Fig. 3a). As shown in Fig. 3b, I3C administration induced the expression of *Cyp1a1* in the neonatal intestinal epithelium of wild-type and *Ahr*[Δlys] but not *Ahr*[-/-] mice, and to a much lesser degree in *Ahr*[ΔIEC] mice, confirming the appropriate activation of the AHR pathway in the neonatal gut. Importantly, I3C supplementation significantly reduced the severity of NEC in wild-type mice (Fig. 3a, c–e) but had a relatively little protective effect in either *Ahr*[ΔIEC] or *Ahr*[-/-] mice

(Fig. 3a, c–e), confirming that AHR activation in the intestinal epithelium by I3C is required for protection against NEC. In an important control, I3C supplementation continued to exert a protective effect when administered to *Ahr*[Δlys] mice, which still express AHR on the intestinal epithelium (Fig. 3a, c–e). For these experiments, the corresponding breast fed controls maintain a normal histology, as shown in Supplementary Fig. 2. Furthermore, in order to gain additional insights into the effectiveness of I3C administration to newborn pups for the prevention of NEC, we performed a dose-response study using doses between

**Fig. 1 Administration of the AHR ligand I3C during pregnancy protects against NEC in the newborn offspring. a** Schematic illustrating the experimental setup in which administration of the AHR ligand I3C to the pregnant mother can be evaluated for effects on the development of NEC in the pups. **b–d** mRNA expression of AHR activation marker *Cyp1a1* in the ileum of pregnant mice (**b**, $n = 3$, 3 mice, $p = 0.0003$), the fetus (**c**, $n = 9$, 8 mice, $p = 0.0317$), and newborn mice at p1 (**d**, $n = 3$, 3 mice, $p = 0.0053$) or neomates at p11 (**e**, $n = 5$, 7 mice, $p = 0.0002$), on control diet (Ctrl) or maternal I3C enriched diet (M-I3C, 25 mg per kg body weight per day). **f–i** representative H&E-stained micrographs (**f**); NEC severity (**g**, $n = 7$, 6 mice, $p = 0.0031$); mRNA expression of *Il6* (**h**, $n = 13$, 13 mice, $p = 0.0329$) and *Tnf-α* (**i**, $n = 13$, 13 mice, $p = 0.0274$) in the terminal ileum of pups with NEC from mothers (i.e., dams) who were fed either a control diet (Ctrl) or a diet rich in I3C (M-I3C, 25 mg per kg body weight per day). **j–l** AHR activation measured as *Cyp1a1* expression in the intestinal epithelial cell line (IEC-6) treated with amniotic fluid (**j**, $n = 11$, 11 wells of cells, $p = 0.0104$), breast milk (**k**, $n = 3$, 3 wells of cells, $p = 0.0002$), and serum (**l**, $n = 13$, 12 wells of cells, $p = 0.0007$) that were harvested from the pregnant mice that were either on the control diet (Ctrl) or I3C enriched diet (M-I3C, 25 mg per kg body weight per day); **m, n** qRT-PCR showing expression of *Il6* (**m**, $n = 7$, 8 mice, $p = 0.0192$) and *Tnf-α* (**n**, $n = 7$, 8 mice, $p = 0.0004$) in the ileum induced by LPS (50 μg per mL for 6 h) injection in neonatal mice, born from mice fed on control diet (Ctrl) or maternal I3C enriched diet (M-I3C, 25 mg per kg body weight per day) during pregnancy. Scale bars in **f**, 100 μm. All data are presented as mean values ± SEM. $*p < 0.05$, $**p < 0.01$, $***p < 0.001$, $p$ values obtained from two-sided $t$ tests. Each dot in graphs represents data from an individual mouse, or an individual well of cell culture.

5–50 mg per kg body weight per day, and identified that doses in the range of 25–50 mg per kg body weight per day significantly activated AHR and protected mice from NEC, justifying the selection of I3C dose of 25 mg per kg body weight per day (Supplementary Fig. 3). Taken together, these findings illustrate that AHR activation on the intestinal epithelium protects against NEC development. We, therefore, next examined the potential mechanisms involved.

**AHR protects against NEC independent of IL-22 and intestinal tight junctions, permeability and IELs.** To investigate potential mechanisms by which AHR activation in the neonatal intestinal epithelium could reduce NEC severity, we next turned to other systems in which AHR activation has been shown to play a protective role. AHR signaling has been shown to critically regulate mucosal Th17 and innate lymphoid type-3 (ILC3) cells, each of which are sources of IL-22[25], a cytokine that is critical for the protective effects of AHR in other models. To investigate whether IL-22 was required for the protection by AHR against NEC, we administered I3C and then induced NEC in *Il22-/-* mice. As shown in Supplementary Fig. 4, I3C supplementation induced the expression of *Cyp1a1* in the intestinal epithelium (Supplementary Fig. 4a) and reduced NEC severity (Supplementary Fig. 4b–d) in *Il22-/-* mice, illustrating that IL-22 was not required for the protective effects of AHR activation on NEC. Consistent with this finding, in the early postnatal period in which NEC is induced, we observed that Th17 cells (Supplementary Fig. 4e) and ILCs (Supplementary Fig. 4f–h) were found to be rare populations in the intestinal mucosa, a finding supported by others[26,27].

We next considered the possibility that AHR activation could regulate tight junction expression and intestinal barrier integrity, as has been shown to occur in older mice[28]. Interestingly, we found no difference between wild-type and *Ahr-/-* mice in the distribution of the tight junction protein ZO-1 in the newborn ileum (Supplementary Fig. 4i). There was also no difference between wild-type and *Ahr-/-* mice on intestinal permeability after oral gavage of labeled dextran at baseline in mice without NEC, or between groups of mice with NEC (Supplementary Fig. 4j), making it unlikely that the protective effects of AHR activation could be attributed to improved barrier integrity. We do note that the presence of NEC does increase permeability over control mice in both groups, suggesting that the NEC model is sufficient to induce an increase in barrier permeability, but insufficient to cause the epithelial and mucosal changes that lead to NEC. Furthermore, when assessing the total proportion of CD45-positive cells and neutrophils in the lamina propria, we determined that NEC significantly increased the number of CD45 cells (Supplementary Fig. 4k), the number of neutrophils (Ly6G + cells, Supplementary Fig. 4l), and the percentage of CD45 cells that are neutrophils (Supplementary

Fig. 4m) in both wild-type and *Ahr-/-* mice. However, although there was a very slight increase of the numbers of CD45 cells and neutrophils in *Ahr-/-* mice compared with wild-type mice at baseline without NEC (Supplementary Fig. 4k, l), the percentage of CD45 cells that were neutrophils did not differ between *Ahr-/-* mice and wild-type mice at baseline (Supplementary Fig. 4m), and there were also no significant differences on the numbers of CD45 cells, the number of neutrophils, and the percentage of neutrophils in CD45 cells in *Ahr-/-* compared with wild-type mice with NEC (Supplementary Fig. 4k–m). These data support the overall findings that the effects of AHR signaling in preventing NEC are not due to broadscale reduction in inflammatory cells.

Prior authors have shown that interepithelial lymphocyte (IEL) subsets may confer protection against NEC and are dependent on AHR ligands for their survival[29,30]. We, therefore, next evaluated the possible role of IELs in the mechanisms by which AHR activation can protect against NEC. As shown in Supplementary Fig. 5a, approximately 90% of the IEL T cells in the newborn mouse intestine were γδ T cells, which led us to focus on this subtype. As shown in Supplementary Fig. 5b, there were no changes in the quantity of IELs between wild-type and *Ahr-/-* pups. Moreover, administration of I3C did not increase the quantity of IELs, and there were also no significant differences on the quantity of IELs between mice with and without NEC (Supplementary Fig. 5c). Next, to directly assess a potential role for IELs in the mechanisms by which I3C protects against NEC, we depleted IELs in mice using a diptheria toxin depletion strategy, and assessed whether I3C administration could still protect against NEC. As shown in Supplementary Fig. 5d, treatment with I3C still induced *Cyp1a1* in the intestinal mucosa in IEL-depleted mice, confirming that AHR activation had occurred within the newborn gut. Significantly, the administration of I3C still protected IEL-depleted pups from NEC (Supplementary Fig. 5e–h), making it unlikely that the effects of I3C can be attributed to IELs.

Based upon these studies, and given that TLR4 plays a critical role in NEC pathogenesis[4,5,11,24], we next turned to the potential role of AHR activation in reducing TLR4 signaling or expression in the neonatal intestinal epithelium.

**AHR activation limits TLR4 signaling and expression in the intestinal epithelium of mice and humans.** To investigate directly whether AHR activation could limit TLR4 signaling in the intestinal epithelium, we next performed studies in primary enteroids derived from both wild-type and *Ahr-/-* mice (Fig. 4a). Treatment of enteroids with I3C induced the expression of *Cyp1a1* in wild-type but not *Ahr-/-* enteroids (Fig. 4b), confirming that AHR is appropriately activated on intestinal epithelial cells. Importantly, I3C significantly reduced the LPS-mediated induction of *Tnf-α* in wild-type but not *Ahr-/-* enteroids (Fig. 4c),

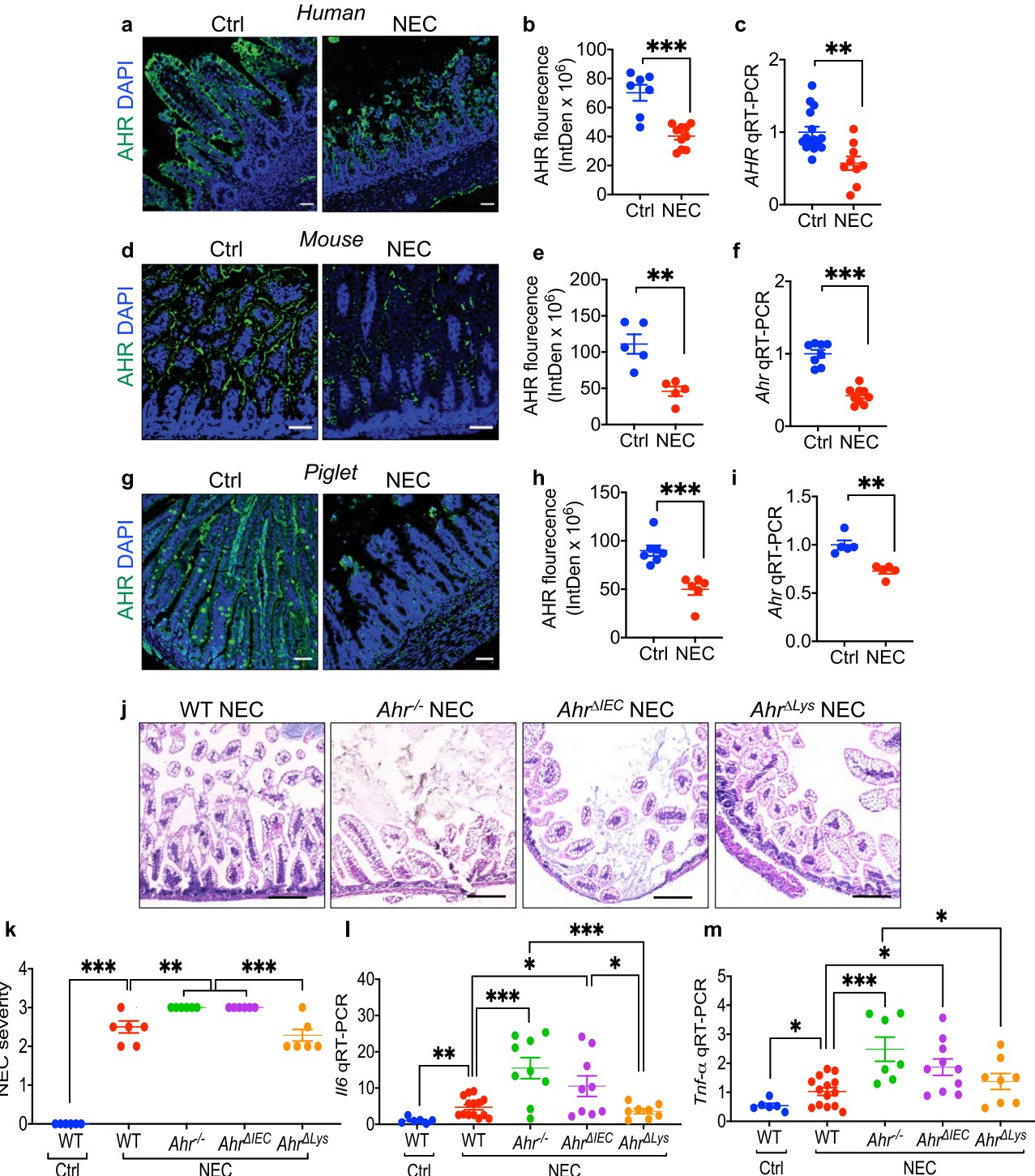

**Fig. 2 AHR expression on the the intestinal epithelium is required for protection against NEC. a**, **b**, **d**, **e**, **g**, **h** Representative confocal images and quantification of fluorescent intensity of AHR immuno-stained ileal sections of human (**a**, **b**, $n = 7$, 10 ileal section, $p < 0.0001$), mouse (**d**, **e**, $n = 5$, 5 ileal section, $p = 0.0025$), piglet (**g**, **h**, $n = 7$, 6 ileal section, $p = 0.0004$) ileal specimens from control (Ctrl) and NEC patients or animals. AHR, green signal; nuclei (DAPI, blue signal). **c**, **f**, **i** Expression of *Ahr* by qRT-PCR in the small intestine of human (**c**, $n = 15$, 9 human specimens, $p = 0.0019$), mouse (**f**, $n = 8$, 9 mice, $p < 0.0001$), and piglets (**i**, $n = 5$, 5 piglets, $p = 0.0010$). **j** H&E-stained representative images showing abnormal NEC histology in mice induced to develop experimental NEC in wild-type (WT), AHR knockout (*Ahr$^{-/-}$*), AHR intestinal epithelial cells knockout (*Ahr$^{\Delta IEC}$*), and AHR myeloid knockout (*Ahr$^{\Delta lys}$*) mice. **k** Quantification of NEC severity ($n = 7$, 7, 7, 7, 7 mice, WT Ctrl vs WT NEC, $p < 0.0001$, WT NEC vs *Ahr$^{-/-}$* NEC $p = 0.0074$, WT NEC vs *Ahr$^{\Delta IEC}$* NEC $p = 0.0074$, *Ahr$^{-/-}$* NEC vs *Ahr$^{\Delta lys}$* NEC $p = 0.0001$, *Ahr$^{\Delta IEC}$* NEC vs *Ahr$^{\Delta lys}$* NEC $p = 0.0001$)· **l**, **m** qRT-PCR expression of *Il6* (**l**, $n = 6$, 14, 7, 9, 8 mice, WT Ctrl vs WT NEC, $p = 0.0048$, WT NEC vs *Ahr$^{-/-}$* NEC $p = 0.0006$, WT NEC vs *Ahr$^{\Delta IEC}$* NEC $p = 0.0268$, *Ahr$^{-/-}$* NEC vs *Ahr$^{\Delta lys}$* NEC $p = 0.0009$, *Ahr$^{\Delta IEC}$* NEC vs *Ahr$^{\Delta lys}$* NEC $p = 0.0415$) and *Tnf-α* (**m**, $n = 6$, 14, 7, 9, 8 mice, WT Ctrl vs WT NEC, $p = 0.0350$, WT NEC vs *Ahr$^{-/-}$* NEC $p = 0.0012$, WT NEC vs *Ahr$^{\Delta IEC}$* NEC $p = 0.0076$, *Ahr$^{-/-}$* NEC vs *Ahr$^{\Delta lys}$* NEC $p = 0.0473$) in the intestinal epithelium. Scale bars in **a**, **d**, **g**, 50 μm. Scale bars in **j**, 100 μm. All data are presented as mean values ± SEM. *$p < 0.05$, **$p < 0.01$, ***$p < 0.001$, $p$ values obtained either from two-sided $t$ tests or one-way ANOVA followed by multiple comparisons. Each dot in graphs represents data from an individual ileal section, human specimen, mouse, or piglet.

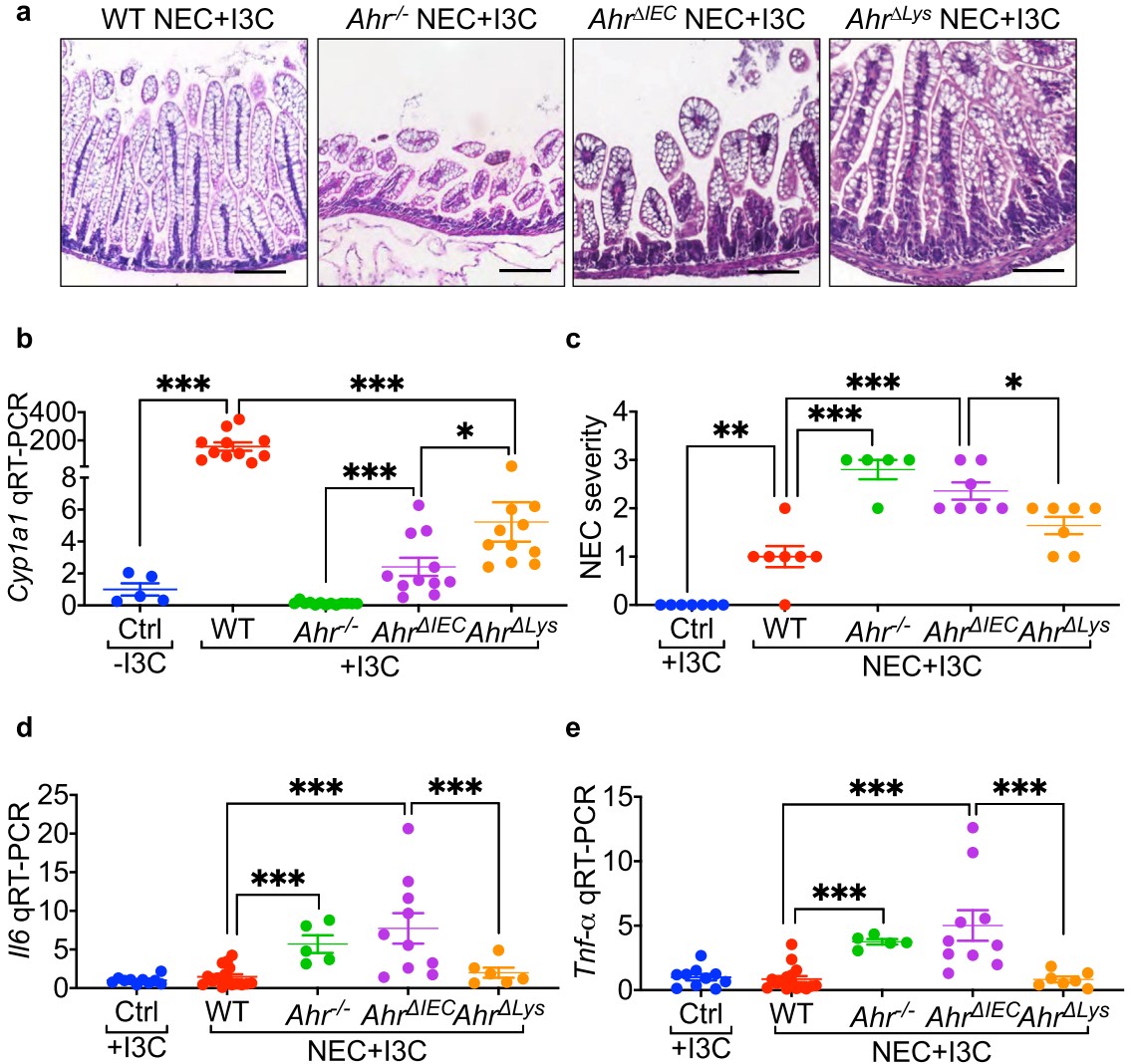

**Fig. 3 Feeding mice the AHR ligand I3C protects against NEC by activating AHR on the intestinal epithelium. a** H&E-stained representative images in ileal sections of newborn mice, supplemented with AHR ligand I3C (25 mg per kg body weight per day for 4 days) and induced to develop experimental NEC. Data showing I3C mediated protection against NEC development only in wild-type (WT) and AHR myeloid knockout (Ahr^ΔLys) mice but not in AHR knockout (Ahr^-/-) and AHR intestinal epithelial cells knockout (Ahr^ΔIEC) mice. **b** Dot graph showing AHR ligand I3C supplementation (25 mg per kg body weight per day for 4 days) produced a multifold induction of AHR activation marker Cyp1a1 in the ileum of wild-type but not in Ahr^-/-, and Ahr^ΔIEC and a moderate increase in Ahr^ΔLys mice (n = 5, 11, 13, 11, 11 mice, Ctrl -I3C vs WT + I3C p < 0.0001, WT + I3C vs Ahr^ΔLys + I3C p < 0.0001, Ahr^-/- + I3C vs Ahr^ΔIEC + I3C p = 0.0002, Ahr^ΔIEC + I3C vs Ahr^ΔLys + I3C p = 0.0104). **c-e** NEC severity (**c**, n = 7, 7, 5, 7, 7 mice, Ctrl +I3C vs WT NEC + I3C p = 0.0020, WT NEC + I3C vs Ahr^-/- NEC + I3C p < 0.0001, WT NEC + I3C vs Ahr^ΔIEC NEC + I3C p < 0.0001, Ahr^ΔIEC NEC + I3C vs Ahr^ΔLys NEC + I3C p = 0.0402) and mRNA levels of pro-inflammatory cytokine Il6 (**d**, n = 10, 16, 5, 10, 6 mice, WT NEC + I3C vs Ahr^-/- NEC + I3C < 0.0001, WT NEC + I3C vs Ahr^ΔIEC NEC + I3C p = 0.0001, Ahr^ΔIEC NEC + I3C vs Ahr^ΔLys NEC + I3C p = 0.0086**)** and Tnf-α (**e**, n = 10, 16, 5, 10, 6 mice, WT NEC + I3C vs Ahr^-/- NEC + I3C p < 0.0001, WT NEC + I3C vs Ahr^ΔIEC NEC + I3C p < 0.0001, Ahr^ΔIEC NEC + I3C vs Ahr^ΔLys NEC + I3C p = 0.0004) in the ileum of control mice without NEC and wild-type, Ahr^-/-, Ahr^ΔIEC, and Ahr^ΔLys mice with NEC with I3C supplementation (25 mg per kg body weight per day for 4 days). Scale bars in **a**, 100 μm. All data are presented as mean values ± SEM. *p < 0.05, **p < 0.01, ***p < 0.001, p values obtained from two-sided t tests or one-way ANOVA followed by multiple comparisons. Each dot in graphs represents data from an individual mouse.

consistent with a reduction in TLR4 signaling in intestinal epithelial cells in vitro. As shown in Fig. 4d, e, I3C treatment significantly reduced LPS-induced translocation of the down-stream transcription factor NF-κB from the cytoplasm to the nucleus in these enteroids, providing an additional measure of TLR4 inhibition.

In seeking to understand whether I3C could inhibit intestinal TLR4 in vivo, we fed wild-type p11 mice a diet rich in I3C, which induced Cyp1a1 expression in their intestinal epithelium confirming AHR activation as expected (Fig. 4f). Importantly, feeding I3C to wild-type p11 mice resulted in significant attenuation of TLR4 signaling, as manifest by reduced LPS-induced Il6 (Fig. 4g)

and Tnf-α (Fig. 4h) expression in the intestinal epithelium, and which was accompanied by reduced expression of Tlr4 in the intestinal mucosa (Fig. 4i). I3C administration did not induce Cyp1a1 (Fig. 4f) and did not reduce either TLR4 signaling (Fig. 4g, h) or Tlr4 expression (Fig. 4i) in Ahr^-/- mice, confirming the specificity of the I3C effect. In seeking to understand how AHR activation could reduce Tlr4 expression, we note that I3C administration significantly upregulated the expression in the ileum of microRNAs that are known to negatively regulate Tlr4 expression, namely miR-146b, miR-223, and let-7i in wild type mice[31], while I3C did not increase the expression of these microRNAs in Ahr^-/- mice (Fig. 4j–l).

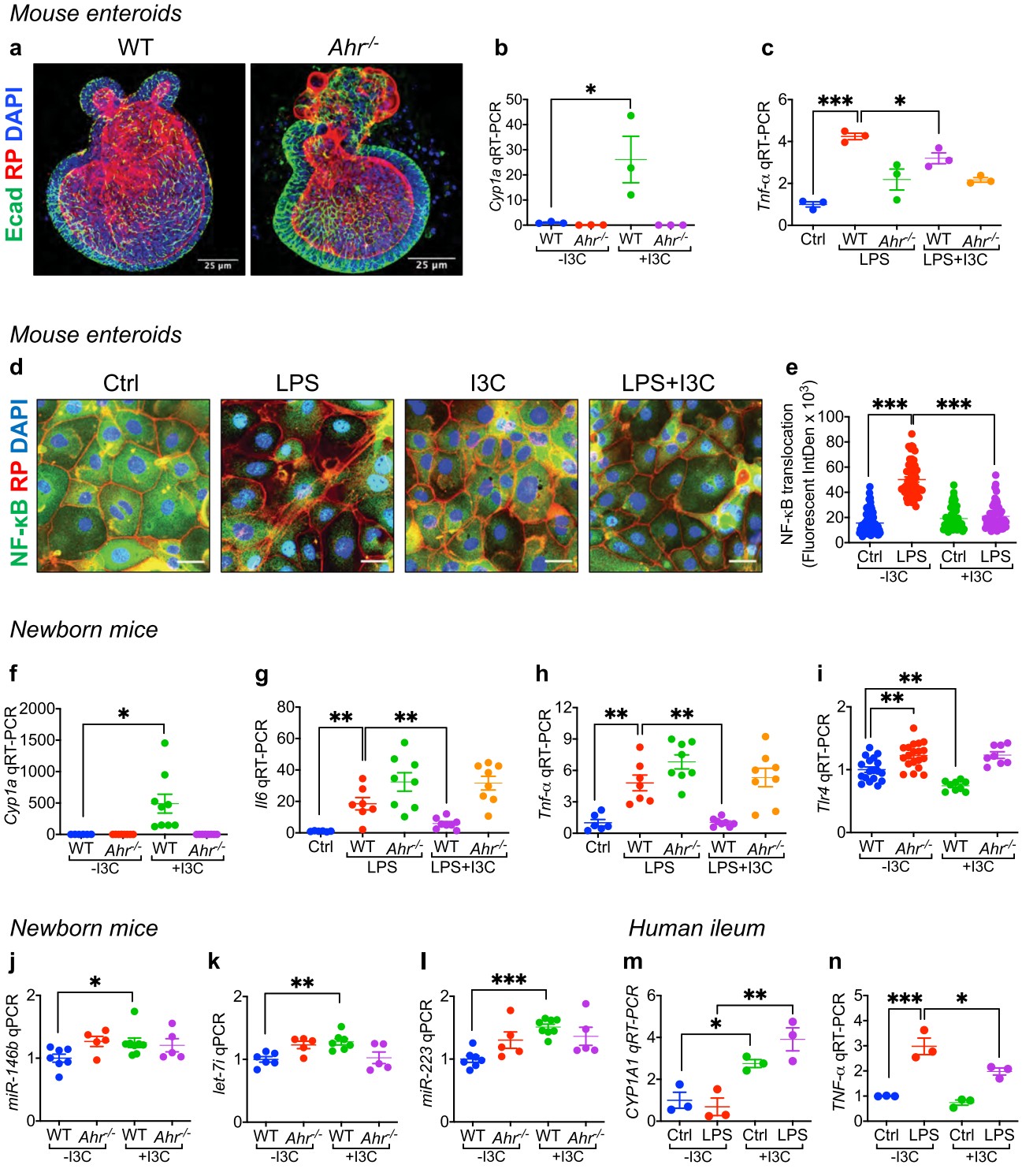

To assess whether AHR activation could reduce TLR4 signaling in human intestine, we next treated intestinal explant cultures derived from freshly resected intestinal samples from premature infants undergoing surgery for NEC, with both LPS and I3C. As shown in Fig. 4m, n, I3C treatment of human intestinal cultures induced the expression of *CYP1A1*, revealing the activation of AHR in human tissue (Fig. 4m), and also reduced TLR4 signaling, as revealed by reduced LPS-induced *TNF-α* expression (Fig. 4n). Taken together, these findings indicate that AHR activation inhibits TLR4 signaling and expression in the intestinal epithelium. Given our findings that maternal–fetal

signaling can regulate NEC, as well as our observation in Fig. 1 that breast milk can induce the AHR downstream gene *Cyp1a1*, we next explored whether AHR signaling could explain in part the protective effects of breast milk for NEC.

**Breast milk reduces TLR4 signaling and prevents NEC via AHR on the newborn intestinal epithelium.** Having shown that the administration of a maternal diet rich that is in AHR ligands can protect NEC in the offspring, we next sought to evaluate whether these findings could extend to breast milk—which is by definition rich in maternal-derived ligands. Specifically, we

**Fig. 4 AHR activation limits TLR4 signaling and expression in the intestinal epithelium of mice and humans. a** Representative confocal images of enteroids harvested from wild-type (WT) and AHR knockout ($Ahr^{-/-}$) mice after 7 days of culture on Matrigel, and stained with the epithelial marker (Ecadherin, Ecad, green signal), actin filaments (Rhodamine Phalloidin, RP, red signal), and nuclei (DAPI, blue signal). **b, c** qRT-PCR showing the expression of *Cyp1a1* (**b**, n = 3, 3, 3, 3 wells of enteroids, WT -I3C vs WT + I3C p = 0.0208) and *Tnf-α* (**c**, n = 3, 3, 3, 3 wells of enteroids, Ctrl vs WT LPS p < 0.0001, WT LPS vs WT LPS + I3C p = 0.0258) in these enteroids under the indicated condition (200 μM I3C pretreatment overnight and then LPS treatment (50 μg per mL) for 4 h). **d, e** Representative confocal images (**d**) and quantification of fluorescent intensity of NF-κB translocation (**e**, n = 81, 48, 48, 50 enteroid cells, Ctrl -I3C vs LPS -I3C p < 0.0001, LPS -I3C vs LPS + I3C p < 0.0001). NF-κB, green signal; actin filaments (Rhodamine Phalloidin, RP, red signal); nuclei (DAPI, blue signal). **f–i** qRT-PCR showing the expression of *Cyp1a1* (**f**, n = 6, 8, 9, 8 mice, WT -I3C vs WT + I3C p = 0.0042), *Il6* (**g**, n = 6, 7, 8, 8, 8 mice, Ctrl vs WT LPS p = 0.0018, WT LPS vs WT LPS + I3C p = 0.0065), *Tnf-α* (**h**, n = 6, 7, 8, 8, 8 mice, Ctrl vs WT LPS p = 0.0036, WT LPS vs WT LPS + I3C p = 0.0017) and *Tlr4* (**i**, n = 18, 19, 9, 8 mice, WT -I3C vs $Ahr^{-/-}$ -I3C p = 0.0013, WT -I3C vs WT + I3C p = 0.0033) in the ileum of newborn mice after treatment with or without I3C (25 mg per kg body weight per day for 4 days). **j, k** qRT-PCR showing the expression of the *Tlr4* regulatory microRNAs, *miR-146b* (**j**, n = 7, 5, 8, 5 mice WT -I3C vs WT + I3C p = 0.0300), *let-7i* (**k**, n = 6, 5, 7, 5 mice WT -I3C vs WT + I3C p = 0.0016) and *miR-223* (**l**, n = 7, 5, 8, 5 mice WT -I3C vs WT + I3C p < 0.0001) in the ileum of WT and $Ahr^{-/-}$ mice in the presence or absence of I3C (25 mg per kg body weight for 24 h). **m, n** qRT-PCR showing expression of *CYP1A1* (**m**, n = 3, 3, 3, 3 wells of human explant culture, Ctrl -I3C vs Ctrl +I3C p = 0.0147, LPS -I3C vs LPS + I3C p = 0.0096) and *TNF-α* (**n**, n = 3, 3, 3, 3 wells of human explant culture, Ctrl -I3C vs LPS -I3C p = 0.0003, LPS -I3C vs LPS + I3C p = 0.0218) in the presence of I3C (200 μM I3C pretreatment for 15 min and then additional 6 h) and LPS (50 μg per mL for 6 h). Scale bars in **a**, 25 μm. Scale bars in **d**, 10 μm. All data are presented as mean values ± SEM. *p < 0.05, **p < 0.01, ***p < 0.001, p values obtained either from two-sided t tests or one-way ANOVA followed by multiple comparisons. Each dot in graphs represents data from an individual well of enteroids culture, an individual enteroid cell of NF-κB staining, an individual mouse, or an individual well of human explant culture.

sought to investigate whether breast milk administration could activate AHR in the intestinal mucosa of the newborn mouse pup, and then reduce TLR4 signaling and prevent NEC. To test this possibility directly, we first harvested enteroids from the ilea of wild-type and $Ahr^{-/-}$ mice, and then treated these enteroids with LPS in the presence or absence of human breast milk. As shown in Fig. 5a, treatment of wild-type enteroids with breast milk activated AHR, as revealed by the induction of *Cyp1a1*, a finding not seen in either saline-treated or $Ahr^{-/-}$ enteroids, and revealing that breast milk is indeed enriched in maternally-derived AHR ligands. Importantly, breast milk reduced TLR4 signaling in enteroids in an AHR-dependent manner (Fig. 5b), as revealed by the finding that breast milk significantly reduced LPS-induced *Tnf-α* expression in wild-type but not $Ahr^{-/-}$ enteroids.

The induction of apoptosis in the intestinal epithelium in response to TLR4 activation is an important feature of mouse and human NEC[4,32]. Given that breast milk administration reduces NEC-induced intestinal injury, we next assessed whether breast milk could prevent TLR4-induced apoptosis in enteroids via AHR activation. As shown in Fig. 5c, d, LPS significantly increased enterocyte apoptosis in wild-type enteroids, and the degree of induction was reduced after treatment with breast milk, a protective effect that was absent in $Ahr^{-/-}$ enteroids (Fig. 5c, d). Breast milk also required AHR to inhibit TLR4 signaling in the newborn intestinal epithelium, as the oral administration of breast milk to mice that had been maintained without oral feeds for 3 h induced the expression *Cyp1a1* (confirming AHR activation) (Fig. 5e) and significantly reduced LPS-induced *Il6* (Fig. 5f) and *Tnf-α* (Fig. 5g) expression in the intestinal mucosa of wild-type but not $Ahr^{-/-}$ mice.

To assess whether the protective effects of breast milk for NEC *required* AHR, we supplemented infant formula with human breast milk and observed a reduction in NEC severity in wild-type mice but almost no protection by breast milk on NEC in $Ahr^{-/-}$ mice, as revealed by the lack of reduction in histology and NEC severity score (Fig. 5h, i), or the expression of *Il6* (Fig. 5j) and *Tnf-α* (Fig. 5k), or *Tlr4* (Fig. 5l). Taken together, these findings reveal that maternally-derived factors secreted into the breast milk can reduce TLR4 signaling and attenuate NEC in the offspring. We therefore finally sought to exploit these observations therapeutically, and thus searched for AHR agonizts that could prevent NEC after maternal administration.

**Identification of the AHR ligand, "A18", which activates AHR and reduces TLR4 signaling in human tissue and prevents NEC in mice when administered during pregnancy.** A major goal in

the NEC field is to identify agents that could be administered during pregnancy to reduce the risk of NEC in the event of premature birth. In order to search for AHR ligands that could be used to prevent or treat NEC when administered during pregnancy, we next screened a clinical compound library containing FDA-approved drugs[33] using an AHR-luciferase reporter intestinal epithelial cell line as described in Methods. Our lead compound, shown in Fig. 6a, is a 369 KDa molecule with the formula $C_{16}H_{14}F_3N_3O_2S$, herein called "A18". The administration of A18 induced luciferase expression in a dose dependent manner in the AHR-luciferase reporter cells (Fig. 6b). A18 also induced the expression of *Cyp1a1* in wild-type IEC-6 enterocytes (Fig. 6c) and in the intestinal epithelium when fed to wild-type but not $Ahr^{-/-}$ newborn mice (Fig. 6d), confirming that A18 signals through AHR. Importantly, feeding A18 to mice significantly reduced LPS-induced *Il6* (Fig. 6e) and *Tnf-α* (Fig. 6f) expression, and reduced *Tlr4* expression in the newborn ileum (Fig. 6g), providing further confirmation of the ability of AHR activation to limit TLR4 signaling in the neonatal gut. A18 administration significantly reduced NEC severity when administered orally to wild-type but not $Ahr^{-/-}$ mice, as manifest by improved histology (Fig. 6h), improved NEC severity score (Fig. 6i), and reduced expression of *Il6* (Fig. 6j) and *Tnf-α* (Fig. 6k). The potential clinical relevance of A18 was found as the ex vivo treatment of human intestine obtained from infants undergoing surgery with A18 significantly induced *Cyp1a1* expression (Fig. 6l), confirming its ability to activate AHR in human tissue, and significantly reduced LPS-induced *Tnf-a* expression (Fig. 6m) and *TLR4* expression (Fig. 6n), consistent with the findings in mice (Fig. 6f, g).

Finally, to assess the ability of in utero A18 administration to modulate the maternal–fetal signaling pathway and reduce NEC, we administered A18 to mice during pregnancy. As shown in Fig. 6o, the administration of A18 during pregnancy induced the expression of *Cyp1a1* in the offspring pup intestine, and also significantly reduced NEC severity as manifest by reduced *Il6* (Fig. 6p) and *Tnf-α* (Fig. 6q) and *Tlr4* expression (Fig. 6r) in the neonatal gut, and improved histology (Fig. 6s) and NEC severity (Fig. 6t). Taken together, these results suggest a surprising role for maternal–fetal AHR signaling in NEC, and raise the possibility that A18 can serve as an agent that can be administered either during pregnancy or postnatally for the prevention and treatment of this devastating disease.

## Discussion

The persistently high mortality of NEC reveals both our lack of a sufficient understanding of its pathogenesis, and an urgency to

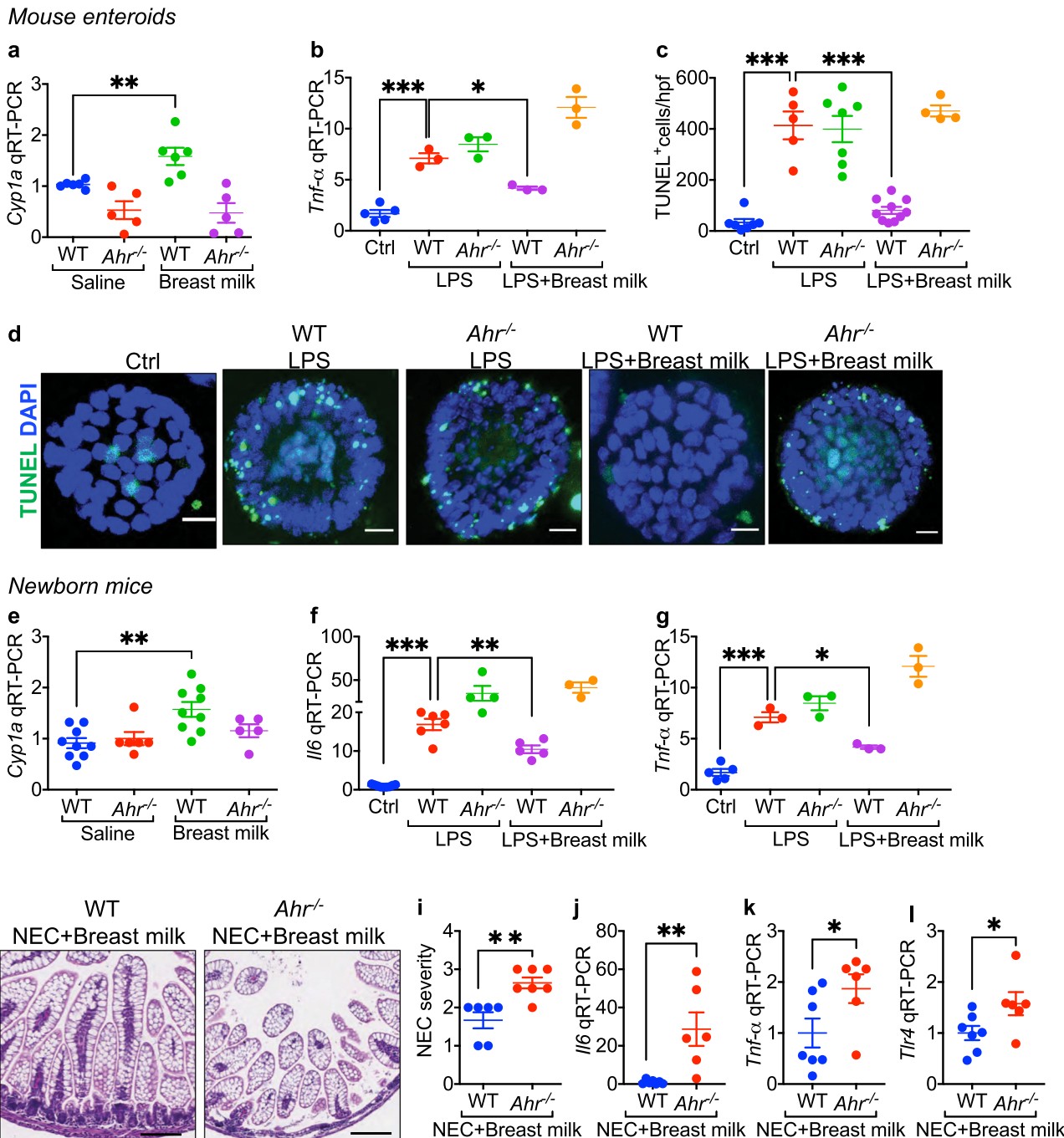

**Fig. 5 Breast milk activates AHR on the intestinal epithelium and protects against experimental NEC in newborn mice. a, b** qRT-PCR showing the expression of *Cyp1a1* (**a**, *n* = 6, 5, 6, 5 wells of enteroids, WT saline vs WT breast milk *p* = 0.0096) and *Tnf-α* (**b**, *n* = 5, 3, 3, 3, 3 wells of enteroids, Ctrl vs WT LPS *p* = 0.0001, WT LPS vs WT LPS + breast milk *p* = 0.0325) in mouse intestinal enteroids from wild-type (WT) and *Ahr^−/−* mice, treated with saline or LPS in the presence or absence of breast milk. **c, d** LPS-induced apoptosis in enteroids measured by TUNEL assay in wild-type and *Ahr^−/−* mouse enteroids as quantified (**c**, *n* = 7, 5, 7, 10, 4 enteroid sections, Ctrl vs WT LPS *p* < 0.0001, WT LPS vs WT LPS + breast milk *p* < 0.0001) and revealed (**d**) by confocal microscopy. TUNEL, green signal; nuclei (DAPI, blue signal). **e–g** qRT-PCR showing the expression of *Cyp1a1* (**e**, *n* = 9, 6, 9 5 mice, WT saline vs WT breast milk *p* = 0.0026), *Il6* (**f**, *n* = 9, 6, 4, 5, 3 mice, Ctrl vs WT LPS *p* = 0.0039, WT LPS vs WT LPS + breast milk *p* = 0.0070) and *Tnf-α* (**g**, *n* = 5, 3, 3, 3, 3 mice, Ctrl vs WT LPS *p* = 0.0001, WT LPS vs WT LPS + breast milk *p* = 0.0325) in wild-type and *Ahr^−/−* mice treated with saline, LPS and/or breast milk as indicated. **h–l** Representative H&E- stained section (**h**), NEC severity score (**i**, *n* = 6, 7 mice, *p* = 0.0023), and the expression of *Il6* (**j**, *n* = 7, 6 mice, *p* = 0.0056), *Tnf-α* (**k**, *n* = 7, 6 mice, *p* = 0.0221) and *Tlr4* (**l**, *n* = 7, 6 mice, *p* = 0.0465) in wild-type and *Ahr^−/−* mice induced to develop NEC in the presence or absence of breast milk as indicated. Scale bars in **d**, 10 μm. Scale bars in **h**, 100 μm. All data are presented as mean values ± SEM. **p* < 0.05, ***p* < 0.01, ****p* < 0.001, *p* values obtained either from two-sided *t* tests or one-way ANOVA followed by multiple comparisons. Each dot in graphs represents data from an individual well of enteroids, section of enteroids, or an individual mouse.

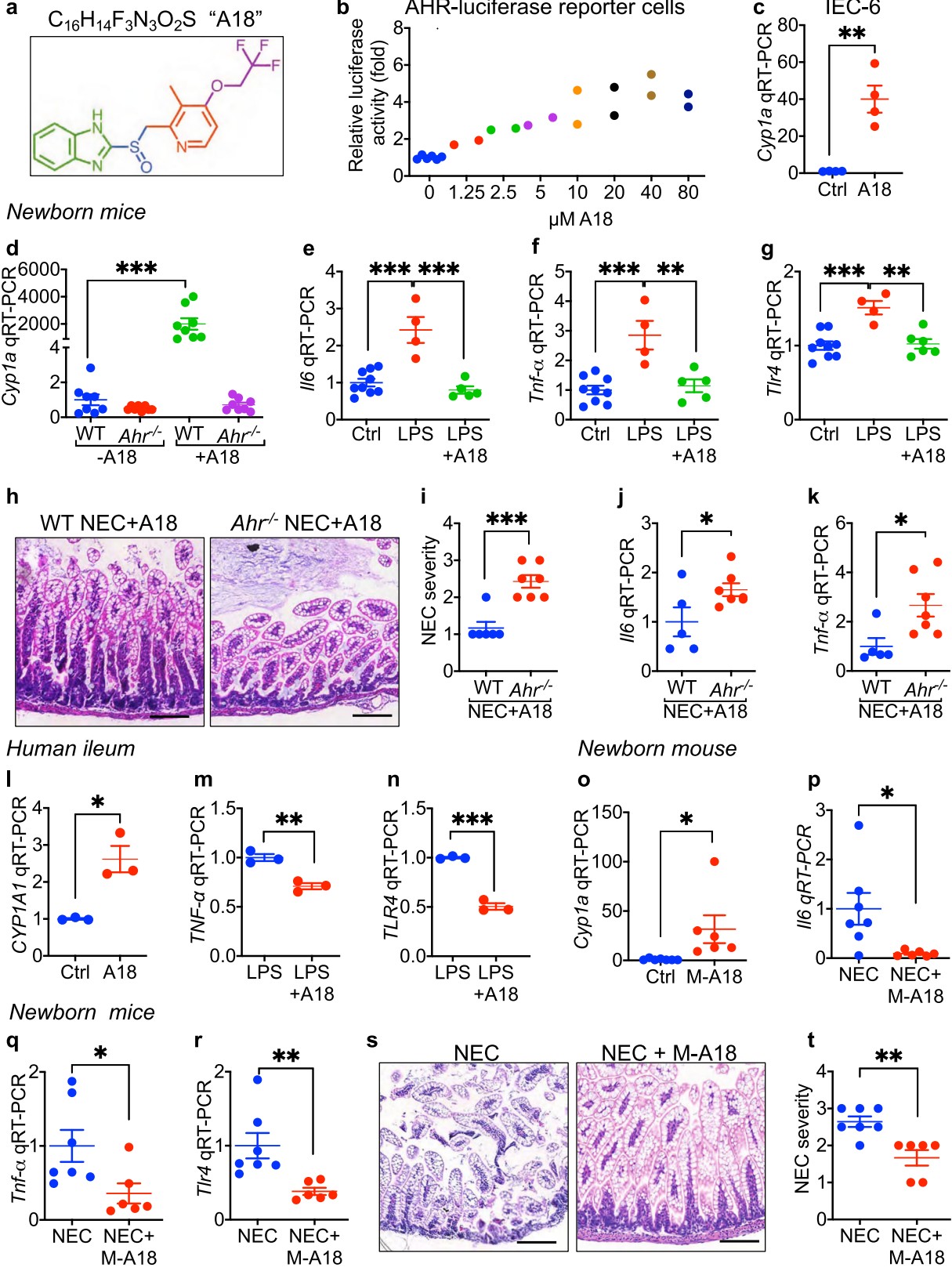

approach the disease differently[34]. In this study, we shed light on the possibility that NEC arises from reduced AHR signaling in the intestinal epithelium of the premature infant, and show that AHR ligands may be passed from mother to infant—both during pregnancy via the fetal circulation, and in the postnatal period through the breast milk—where they can then attenuate the severity of this disease. The mechanism by which AHR activation in the intestinal epithelium attenuates NEC severity involves a reduction in signaling and expression of the innate immune receptor TLR4, whose expression is elevated in the premature bowel as compared with the full term bowel[5,9], and whose activation on the intestinal epithelium we have shown to be critical

**Fig. 6 The AHR agonist "A18" activates AHR on the intestinal epithelium, reduces TLR4 signaling, and protects against experimental NEC in newborn mice. a** Molecular formula of A18. **b** Dose-response curve of A18 for luciferase activity in AHR-reporter IEC-6 cells ($n = 6, 2, 2, 2, 2, 2, 2$ wells of cells, 0 vs 2.5 $p = 0.0390$, 0 vs 5 $p = 0.0081$, 0 vs 10 $p = 0.0006$,). **c** qRT-PCR showing expression of *Cyp1a1* in IEC-6 cells treated with A18 (20 μM for 6 h) ($n = 4, 4$, wells of cells, $p = 0.0018$). **d–g** qRT-PCR showing expression of *Cyp1a1* (**d**, $n = 8, 8, 8, 8$ mice, WT - A18 vs WT + A18 $p < 0.0001$), *Il6* (**e**, $n = 9, 4, 5$ mice, Ctrl vs LPS $p < 0.0001$, LPS vs LPS + A18 $p < 0.0001$), *Tnf-α* (**f**, $n = 9, 4, 5$ mice, Ctrl vs LPS $p = 0.0003$, LPS vs LPS + A18 $p = 0.0017$) and *Tlr4* (**g**, $n = 9, 4, 5$ mice, Ctrl vs LPS $p = 0.0004$, LPS vs LPS + A18 $p = 0.0011$) in the ileum of wild-type (WT) but not AHR knockout (Ahr$^{-/-}$) mice. **h–k** Representative histological H&E of the terminal ileum (**h**), NEC severity score (**i**, $n = 6, 7$ mice, $p = 0.0003$), expression of *Il6* (**j**, $n = 5, 7$ mice, $p = 0.0484$) and *Tnf-α* (**k**, $n = 5, 7$ mice, $p = 0.0222$) in newborn mice with NEC after administration of A18 (300 mg per kg body weight per day for 4 days). **l–n** qRT-PCR showing expression of *CYP1A1* (**l**, $n = 3, 3$ wells of human explant culture, $p = 0.0106$), *TNF-α* (**m**, $n = 3, 3$ wells of human explant culture, $p = 0.0035$) and *TLR4* (**n**, $n = 3, 3$ wells of human explant culture, $p = 0.0001$) in freshly harvested human ileum obtained at surgery treated with LPS (50 μg/mL for 6 h) and/or A18 (20 μM A18 pretreatment for 15 min and then additional 6 h). **o–t** qRT-PCR showing expression *Cyp1a1* (**o**, $n = 7, 6$ mice, $p = 0.0377$) in the neonatal ileum after maternal administration of A18 (300 mg per kg body weight per day for 4 days), expression of *Il6* (**p**, $n = 7, 6$ mice, $p = 0.0254$), *Tnf-α* (**q**, $n = 7, 6$ mice, $p = 0.0346$) and *Tlr4* (**r**, $n = 7, 6$ mice, $p = 0.0081$), ileal H&E of the newborn (**s**) and NEC severity (**t**, $n = 7, 6$ mice, $p = 0.0023$) after NEC induction in the absence or presence of maternal A18 (M-A18, 300 mg per kg body weight per day for 4 days). Scale bars in **h**, **s**, 100 μm. All data are presented as mean values ± SEM. *$p < 0.05$, **$p < 0.01$, ***$p < 0.001$, $p$ values obtained either from two-sided $t$ tests or using one-way ANOVA followed by multiple comparisons. Each dot in graphs represents data from an individual of cell culture, an individual mouse, or an individual well of human explant culture.

for NEC development[4,5,27]. The therapeutic potential of the current findings was revealed by our identification of the AHR ligand A18 to prevent NEC when fed to mice during pregnancy, and by showing its ability to reduce TLR4 signaling in human bowel ex vivo. In view of the fact that the absence of breast milk is a major risk factor for NEC[35–38], these findings suggest the possibility that the development of NEC may reflect impaired AHR signaling in the neonatal intestine, and also show that strategies to enhance the delivery of AHR ligands either directly to the neonate, or secondarily through the mother, may offer new strategies for the prevention or treatment of NEC.

The current studies reveal that AHR signaling on leukocytes plays a minor role in the protection against NEC, in favor of AHR signaling on the intestinal epithelium. This observation is distinct from a separate body of work that reveals that the activation of AHR on leukocytes is required for the maintenance of Th17 and ILC3 cells in the intestinal mucosa[19,39], leading to AHR-mediated protection from colitis through IL-22 release[40]. Differences between those studies and our own may lie in the fact that Th17, IEL, and ILC3 cells are rare in the neonatal intestinal mucosa at the time points in which NEC develops[26,27]. It is noteworthy that intestinal AHR signaling has also been linked to the release of antimicrobial peptides[19] and to the increased differentiation of secretory cells[41], processes which are both downstream of TLR4 activation in the gut[5,42], and thus consistent with the current work. The current work also calls attention to the unique features of NEC, including differences in the location of disease along the gastrointestinal tract (NEC is an ileal disease while colitis affects the colon), age (NEC occurs in the newborn period whereas these other models occur in older mice), as well as differences in microbial ligands which may be present in models of colitis versus NEC, and which could activate AHR. The current studies do not completely exclude a role for AHR signaling on myeloids cells in the pathogenesis of NEC, but rather they argue that NEC protection in response to AHR ligands requires AHR signaling on the intestinal epithelium. Taken together, these findings help provide a link between AHR, dietary factors and the unique immune characteristics of the neonatal gut[21].

One of the most translationally relevant findings of the current study is the identification of an AHR ligand that can prevent NEC, either when offered orally during pregnancy (Fig. 6p–t), or when administered directly to the neonatal pups (Fig. 6h–k). As revealed from its chemical structure in Fig. 6a, the AHR ligand A18 belongs to the benzimidazole class of proton pump inhibitors, which also includes omeprazole[43]. These FDA-approved drugs are currently in clinical use for hyperacidity diseases, and could be conceivably repurposed for NEC, just as they have been

repositioned as anticancer therapeutics[44]. The identification of the heterocyclic A18 as a potent AHR substrate reflects the broad range of hydrophobic structural motifs that bind to the AHR[45]. In seeking to understand the chemical mechanism of AHR binding so as to develop analogs for clinical use, we note that the indole nucleus is a privileged ligand for AHR[46], suggesting that the benzimidazole core of A18 is the main pharmacophore for AHR activation. Major limits of A18 for clinical use in preventing NEC include the fact that while A18 is rapidly absorbed and approximately 97% bound in human plasma, it is extensively metabolized by CYP3A4 and CYP2C18, and plasma elimination half-life is only approximately 2 h in humans[47], leaving much room for further improvement. I3C itself has limitations, including toxicity to the male urogenital tract when used at doses of 100 mg per kg in utero, higher than the doses used in the current set of studies[48]. A18 will need to be modified in order to enhance its AHR signaling and clinical efficacy against NEC.

The current findings have several potential points of impact on clinical medicine, by now showing that NEC may not only be a disease of the postnatal period, but may also reflect impaired signaling in the in utero environment through AHR. We also now provide for the opportunity to interfere with the molecular pathways that lead to the development of NEC through the intra-partum oral delivery of AHR ligands including I3C or A18. If confirmed in clinical studies, these findings may offer the unique ability to intervene in the setting of premature labor, by administering an AHR ligand that could serve to protect the gut and reduce the risk of NEC development in the neonate. In support of the possible success of a strategy in which antenatal delivery of AHR ligands may protect the neonate, it is noteworthy that pregnant women who adhered to a Mediterranean diet—which is rich in AHR ligands—were found to have significantly less NEC than mothers who did not adhere to a Mediterranean diet[49]. Taken in aggregate, the current findings offer the prospect of a renewed approach to NEC, and perhaps finally to a chance to alter the trajectory of this devastating disease.

## Methods

**Animal experiments.** All mice were purchased from the Jackson Laboratory. To generate tissue specific knockouts of *Ahr* from either epithelial cells (Ahr$^{ΔIEC}$) or leukocytes (Ahr$^{ΔLys}$), mice harboring a floxed allele of Ahr$^{fx}$ (Ahrtm3.1Bra/J) were bred with transgenic mice expressing Villin-Cre (*B6.Cg-Tg(Vil1-cre)997Gum/J*) and LysM-Cre (*B6.129P2-Lyz2tm1(cre)Ifo/J*) respectively. To generate global knockouts of *Ahr* (Ahr$^{-/-}$), Ahr$^{fx}$ mice were bred with transgenic mice expressing global Cre under the transcriptional control of a human cytomegalovirus minimal promoter CMV-Cre (B6.C-Tg(CMV-cre)1Cgn/J). The homozygous *Il22*$^{Cre}$ mice (*C57BL/6-Il22tm1.1(icre)Stck/J*), in which the presence of iCre abolishes expression of *Il22*, served as *Il22*$^{-/-}$ mice[50]. ROSA-DTA mice (B6.129P2-Gt(ROSA)26Sor$^{tm1(DTA)Lky}$/J) were bred with TCRδ$^{CreER}$ mice (B6.129S-Tcrd$^{tm1.1(cre/ERT2)Zhu}$/J) to generate

ROSA-DTA/ TCRδ[CreER] mice, in which the TCRγδ IELs can be depleted using tamoxifen. For flow cytometry experiments, Rorγt[GFP] mice (B6.129P2(Cg)-Rorc[tm2Litt]/J) were bred with Ahr[-/-] to generate Foxp3[GFP]; Ahr[-/-] and Rorγt[GFP]; Ahr[-/-] reporter mice, respectively. All mice were housed in a specific pathogen free environment (ambient temperature between 20 and 25 °C, humidity between 30 to 70%) on a 12-hour-light/12-hour-dark cycle with free access to water and standard rodent chow (Teklad global 18% protein rodent diets, Envigo) except otherwise specified.

All mouse and piglet experiments were approved by the Johns Hopkins University Animal Care and Use Committee (MO20M276 for mice and SWl 8M206 for piglets). We have complied with all relevant ethical regulations for animal testing and research have been complied. The genotyping primers are listed in Supplementary Table 1.

**Cell culture**. IEC-6 enterocytes were obtained from ATCC, and maintained in Dulbecco's Modified Eagle's Medium medium containing 10% fetal bovine serum, 40 µg per mL insulin (Gibco), 100 units per mL penicillin, and 100 µg per mL streptomycin.

**Discovery of AHR ligands for the prevention or treatment of NEC in mice**. The intestinal epithelial cells line IEC-6 (ATCC CRL-1592) was stably transduced with lentiviral particles containing Cignal Lenti XRE Reporter (luc) (Qiagen). Cells were treated serially with individual chemicals contained within the Johns Hopkins Drug Library (JHDL), which contains a series of FDA-approved drugs[33] (kindly provided by Dr. Jun O. Liu, Johns Hopkins University) at 10 µM for 24 h, and the luciferase activity was quantified using the SpectraMax M3 (Molecular Devices). Hits were first validated for activation of AHR signaling in vitro by incubating IEC-6 cells with 20 µM A18 for 6 h followed by measuring the induction of the AHR activation reporter Cyp1a1 by qRT-PCR, and then validated for activation of AHR signaling in vivo by oral gavage into neonatal C57/Bl6 mice (p11) followed by measuring the expression of Cyp1a1 by qRT-PCR in the intestinal mucosa 24 h later. Our lead compound from these studies is herein labeled "A18".

**Induction of NEC in mice**. Experimental NEC was induced in a well validated and reproducible model in 7-day-old mice of either gender, which were randomly divided into control and test groups[22,23,27], by gavage feeding newborn mice with formula containing Similac Advance infant formula (Abbott Nutrition): Esbilac (PetAg) canine milk replacer, 2:1 ratio, which was supplemented with enteric bacteria made from a stock created from a specimen obtained from an infant with surgical NEC five times per day. Additionally, the mice were subjected to hypoxia (5% $O_2$–95% $N_2$) for 10 min in a hypoxia chamber (Billups-Rothenberg) twice daily for 4 days. The AHR ligands I3C and A18 were administered by oral gavage during the induction of NEC at the dose of 25 mg per kg body weight per day and 300 mg per kg body weight per day, respectively. To test the protective effect of breask milk on AHR signaling a NEC, human breast milk that was obtained from a single donor (Innovative Research) was supplemented to the formula at the final concentration of 5%. Age-matched breast milk-fed mouse pups were used as healthy controls. Evaluation of ileal histology and expression of pro-inflammatory cytokines by qRT-PCR at a fixed point in the terminal ileum 2 cm proximal to the cecum, were used to determine the disease severity. Leica Application Suite X v3.4.2.18368 software was used to take images for H&E staining.

**Induction of NEC in piglets**. To induce NEC in piglets, timed-pregnant White Yorkshire (Yorkshire x Landrace) sows were obtained from Oak Hill Genetics, and piglets were delivered prematurely via cesarean section at ~95% gestation as we have described[51], in a modification of the work of Sangild et al[52]. NEC was induced in piglets of either gender, which were randomly divided into control and test groups, by gavage formula at 15 mL per kg every 3 h (120 mL per kg body weight per day) for 4 days (n = 7) of the following (per liter): Pepdite Junior (93.9 g; Nutricia), MCT Oil (38.3 g, USP grade; Now Foods), whey protein isolate (56 g, Now Foods), and 837 g water, which was supplemented with enteric bacteria made from a specimen obtained from an infant with surgical NEC.

**Mice endotoxemia model**. Endotoxemia was induced in neonatal mouse pups (p11) of either gender, which were randomly divided into control and test groups, by administering 5 mg per kg LPS via intraperitoneal injection, and ileal samples were harvested 6 h after LPS treatments. I3C or A18 were given through oral gavage daily 3 days prior to the LPS injection at the dose of 25 mg per kg body weight per day and 300 mg per kg body weight per day, respectively.

**AHR induction in pregnant mice**. For activation of AHR during pregnancy in mice, wild type mice were fed an AHR ligand-free diet[53] (AIN-76A, Bio Serv), and I3C and A18 were administered by oral gavage at the dose of 25 mg per kg body weight per day and 300 mg per kg body weight per day, respectively. I3C and A18 were administered to the pregnant mother daily until the offspring were studied further for experimental NEC or endotoxemia as above.

**Harvest and culture of enteroids from mouse ileum**. Primary intestinal crypt cultures (enteroids) were generated from the ileum of neonatal (p7–p11) wild-type, Ahr[-/-], Ahr[ΔIEC], and Ahr[ΔLys] mice as described[32] and maintained in Matrigel (Corning). The enteroids were digested and passed using TrypLE Express (Gibco) weekly, and used between passage 3 and 10 for all experiments. The enteroids were pre-treated with I3C (200 µM, overnight), A18 (20 µM, overnight), or human breast milk (100 µl per mL, Innovative Research, prepared by 5-minute-cenrifugation at 12,000xg and then filtration of supernatant through a 0.22 µm filter) and then treated with LPS (50 µg per mL) for 4 h for further analysis.

**Human ileal sample collection and explant culture**. De-identified human ileal samples were collected during surgery for NEC or at the time of stoma closure, and the Office of Human Subjects Research Review Boards at Johns Hopkins University approved the collection and use of the samples for the study and waived the informed consent (IRB00094036). The IRB waived a requirement to obtain informed consent as the intestinal tissue was discarded, and was obtained during the course of a surgical procedure that was not affected by the study, and because no demographic information was collected, there was no risk to patients, and immortalized stem cells were not established. For RNA isolation and qRT-PCR analysis, fresh samples were snap-frozen in liquid nitrogen immediately. For human explant culture, fresh ileum samples from NEC patients or patient undergoing stoma re-anastomosis were washed with sterile phosphate-buffered saline containing gentamycin (5 µg per mL), minced into 2- to 4-mm diameter pieces, and then cultured in Dulbecco's modified Eagle growth medium supplemented with 10% fetal bovine serum, 4 µg per mL human recombinant insulin, and 100 µg per mL Primocin. Human ilea explant cultures were then pre-treated with 200 µM I3C or 20 µM A18 for 15 min, and then with 50 µg per mL LPS for 6 h, then processed for total RNA isolation followed by qRT-PCR.

**Immunofluorescence staining**. Five micro meter tissue sections from mouse, piglet, and human intestine were rehydrated, heated in 10 mmol/L citric acid buffer for antigen retrieval, permeabilized with 0.1% Tween-20, probed with primary antibodies (1:200 dilution) overnight at 4 °C, probed with secondary antibody and 4′,6-diamidino-2-phenylindole (DAPI, Biolegend) for 1 h at room temperature, and then mounted in Gelvatol mounting media (Sigma-Aldrich) for imaging. To assess apoptosis, samples were incubated with terminal deoxynucleotidyl transferase dUTP nick end labeling (TUNEL) detection solution (In Situ Cell Death Detection Kit, Roche) as per the manufacturer's instructions. Slides were incubated with the nuclear marker DAPI, mounted using Gelvatol solution prior to imaging using a Nikon Eclipse Ti Confocal microscope. NIS-Elements AR v4.10.01 software was used to take images for IF staining.

For immunofluorescence staining of enteroids, cells were cultured in 10 µL Matrigel on chamber slides. For NF-κB nuclear translocation assay, mouse enteroids were cultured on Laminin-coated chamber slides, and were pre-treated with 200 uM I3C for 30 min followed with 100 ug per mL LPS treatment for 2 h. After 20 min fixation in 4% PFA at room temperature, the enteroids were washed with PBS, probed with primary antibodies (1:200 dilution) overnight at 4 °C, probed with secondary antibody and DAPI for 1 h at room temperature, and then mounted in Gelvatol mounting media for imaging.

Antibodies are: goat-anti AHR (clone M-20, Catalog no. sc-8089, Santa Cruz), goat-anti Ecadherin (Catalog no. AF748, R&D Systems), rabbit-anti ZO-1 (Catalog no. 40-2200, Thermo Fisher Scientific), and mouse-anti NF-κB p65 (clone F-6, Catalog no. sc-8008, Santa Cruz). Rhodamine Phalloidin (Catalog no. R415, Thermo Fisher Scientific) was used as F-actin probe.

**RNA isolation, cDNA synthesis, quantification of mRNA and miRNA**. Total RNA was isolated using the RNeasy mini kit (Qiagen) and complementary DNA was synthesized from 0.5 µg RNA using QuantiTect Reverse Transcription kit (Qiagen) following the manufacturer's protocols. The mRNA quantification was performed on the Bio-Rad CFX96 Real-Time System (Bio-Rad) using iTaq™ universal SYBR® Green supermix (Bio-Rad) and Bio-Rad CFX Manager 3.1 software was used to collect data from qRT-PCR. The relative mRNA expression levels were normalized against the expression of the housekeeping gene ribosomal protein lateral stalk subunit P0 (Rplp0). The primers are listed in Supplementary Table 2.

For microRNA isolation, freshly harvested, snap-frozen ileal tissue was subjected to total RNA isolation using the RNeasy plus universal mini kit (Qiagen) and miRNA was quantified using miScript PCR starter kit (Qiagen) following the manufacturer's protocols. The microRNA quantification was performed on the Bio-Rad CFX96 Real-Time System (Bio-Rad) and Bio-Rad CFX Manager 3.1 software was used to collect data. The relative miRNA expression levels were normalized against the expression of housekeeping miRNA miR-191. The microRNA primers are listed in Supplementary Table 3.

**Isolation of IELs and LP cells, and flow cytometry**. IELs were isolated from the new born mouse small intestine according to the methods of Sheridan et al[54]. In brief, the mesentery was removed from the freshly isolated ileum, and the bowel was then opened longitudinally and cut into 0.5 cm pieces, and incubated in HBSS containing 10% fetal bovine serum, 10 mM HEPES and 1 mM dithioerythritol (Sigma-Aldrich) at 37 °C for 20 min with agitation at 180 rpm. After filtration

through a 70 μm cell strainer, IELs were collected between the interface of 40 and 60% discontinuous Percoll in preparation for flow cytometry.

LP cells were isolated from the newborn mouse ileum according to the methods of Hepworth et al[55]. In brief, the mesentery was removed from the freshly isolated ileum, and the bowel was then opened longitudinally and cut into 1-cm pieces, incubated in PBS containing 5% fetal bovine serum, 1 mM dithioerythritol (Sigma-Aldrich) and 1 mM EDTA at 37 °C for 20 min with agitation at 180 rpm. After filtration through a 70-μm cell strainer, the remaining tissue was finely minced with scissors, and incubated in RPMI containing 2% fetal bovine serum, 0.5 mg per mL collagenase/dispase (Sigma-Aldrich), and 0.02 mg per mL DNase (Sigma-Aldrich) at 37 °C for 40 min with agitation at 180 rpm. After filtration through sequential 70 and 40 μm cell strainers, the lamina propria leukocytes were collected between the interface of 40 and 60% discontinuous Percoll in preparation for flow cytometry.

Single-cell suspensions were washed using ice-cold FACS buffer (PBS, 1% BSA, 0.01% NaN3) and incubated with rat anti-CD16/CD32 (clone 93, Catalog no. 101320, BioLegend) to block Fc receptor binding (1.0 μg per 10⁶ cells in 100 μl volume, 20 min, 4 °C) on mouse cells. Cells were pelleted by centrifugation and resuspended (400xg for 5 min) in optimal concentrations of fluorochrome-conjugated antibodies in ice-cold FACS buffer to stain surface molecules. The dead cells were stained using Fixable Viability Violet Dye (Thermo Fisher) in PBS. Intracellular staining was performed using the Foxp3 buffer set (Biosciences). After washing, the samples were analyzed on a BD LSRII flow cytometer for ILCs or BD Accuri™ C6 Plus flow cytometer. FACSDiva v8.0.2 software was used to collect data from BD LSRII flow cytometer, BD CSampler v1.0.264.21 Software was used to collect data from BD Accuri™ C6 Plus flow cytometer, and data analysis was performed using FlowJo software (v10.6.1). The fluorochrome-conjugated antibodies used in this study include: rat anti-mouse CD90.2 Alexa Fluor® 700 (clone 30-H12, Catalog no. 105319, BioLegend), armenian hamster anti-mouse CD3e PerCP-Cyanine5.5 (clone 145-2C11, Catalog no. 16-0031-81, eBioscience), rat anti-mouse CD5 PerCP-Cyanine5.5 (clone 53-7.3, Catalog no. 45-0051-80, eBioscience), rat anti-mouse CD45R (B220) PerCP-Cyanine5.5 (clone RA3-6B2, Catalog no. 45-0452-80, eBioscience), armenian hamster anti-mouse CD11c PerCP-Cyanine5.5 (clone N-418, Catalog no. 45-0114-80, eBioscience), rat anti-mouse CD11b PerCP-Cyanine5.5 clone (M1/70, Catalog no. 45-0112-80, eBioscience), rat anti-mouse Gata-3 PE-Cyanine7 (clone TWAJ, Catalog no. 25-9966-41, eBioscience), mouse anti-mouse T-bet eFluor® 660 (clone 4B10, Catalog no. 50-5825-80, eBioscience), rat anti-mouse ROR gamma (t) PE (clone B2D, Catalog no. 12-6981-80, eBioscience), rat anti-mouse EOMES Alexa Fluor® 488 (clone Dan11mag, Catalog no. 53-4875-80, eBioscience), rat anti-mouse CD4 APC (clone RM4-5, Catalog no. 561091, BD Biosciences), rat anti-mouse CD45 PerCP-Cyanine5.5 (clone I3/2.3, Catalog no. 147705, BioLegend), rat anti-mouse CD19 Alexa Fluor® 700 (clone 1D3, Catalog no. 56-0193-80, eBioscience), rat anti-mouse CD3 APC (clone 17A2, Catalog no. 17-0032-80, eBioscience), armenian hamster anti-mouse TCR gamma/delta PE (clone GL-3, Catalog no. 12-5711-81, eBioscience), armenian hamster anti-mouse TCR beta FITC (clone H57-597, Catalog no. 11-5961-81, eBioscience), rat anti-mouse CD45 PE (clone I3/2.3, Catalog no. 147711, BioLegend), rat anti-mouse CD11b APC (clone M1/70, Catalog no. 101211, BioLegend), and rat anti-mouse Ly6G FITC (clone 1A8-Ly6g, Catalog no. 11-9668-80, eBioscience). Gating strategies for flow cytometry analysis were shown in Supplementary Fig. 6.

**Intestinal permeability assay**. Pups were gavaged with 500 mg per kg of fluorescein isothiocyanate (FITC)–conjugated dextran (4 kDa, Sigma-Aldrich). Blood was collected from the orbital sinus under isofluorane anesthesia 3 h later, and the serum fluorescence was measured using the SpectraMax M3 spectrophotometer (Molecular Devices). SoftMax Pro v6.4.2 Software was used to collect data from the luciferase assay and FITC-dextron assay.

**Isolation of murine peritoneal cells**. Immediately after euthanization, 1 mL of cold DMEM/F12 containing 10% FBS was injected to the peritoneum of p7 wild-type, $Ahr^{-/-}$, $Ahr^{\Delta IEC}$, and $Ahr^{\Delta Lys}$ mice, and then the peritoneal fluid was collected. The peritoneal exudate cells were seeded into cell culture plates. After 4 h incubation at 37 °C, the suspended cells were removed and the adherent cells, which are predominantly peritoneal macrophages[56], were treated with I3C (200 μM, overnight) and assessed for the expression of $Ahr$ and $Cyp1a1$ by qRT-PCR.

**Statistics and Reproducibility**. All data are presented as mean values ± SEM, and analyzed for statistical significance either from two-sided $t$ tests or one-way ANOVA followed by multiple comparisons using GraphPad Prism 9 (GraphPad). A $p$ value of less than 0.05 was considered statistically significant as indicated. *$p$ < 0.05, **$p$ < 0.01, ***$p$ < 0.001. Graphs show individual dots for each well of cells, well of enteroids, mouse, piglet, or human sample. All experiments were repeared independently with similar results at least three times.

**Reporting summary**. Further information on research design is available in the Nature Research Reporting Summary linked to this article.

## Data availability

All relevant data are available from the corresponding authors upon request, and reagents can be made available with a Material Transfer Agreement between the requesting institution and Johns Hopkins University. Source data are provided with this paper.

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

## Acknowledgements

The authors thank Dr. Jun O. Liu from Johns Hopkins University for providing the Johns Hopkins Drug Library. The authors also would like to thank Timothy H. Phelps from Johns Hopkins University Department of Art as Applied to Medicine for assistance with the illustration in Fig. 1. D.J.H. is supported by R01GM078238 and R01DK117186 from the National Institutes of Health. M.L.K. is supported by T32 DK00771322 from the National Institutes of Health. P.L. was supported by the Research Fellowship for Early Career Faculty Investigators from the North American/United States Shock Society.

## Author contributions

P.L., C.P.S., and D.J.H. conceived and designed the study. P.L., Y.Y., W.B.F., S.W., Q.Z., H.J., M.L.K., A.G.S., M.S., T.P.J., and C.P.S. performed the experiments and data generation. P.L. and C.P.S. performed the data analysis. P.W. carried out the drug screen consultation. D.J.H. wrote the original draft. P.L. and C.P.S. prepared the original figures. D.J.H., M.L.K. and P.L. acquired funding for this study. All authors reviewed and approved the final manuscript.

## Competing interests

D.J.H., P.W., C.P.S., and P.L. have filed a patent application for the use of AHR agonists in the prevention and treatment of NEC. The remaining authors declare no competing interests.
