## [Peer Review File · Nature Communications]

Reviewers' Comments:

Reviewer #1:

Remarks to the Author:

This is an interesting and important study evaluating the effects of AHR stimulation in pregnant mice on the development of a necrotizing enterocolitis (NEC) model with some correlative measurements of AHR expression in humans and piglets with NEC. On the whole, the effect of supplementation with the AHR pro-ligand I3C during pregnancy and the downstream effects on the newborn mice is convincingly demonstrated, suggesting that the AHR status of the mother has a strong impact on intestinal health of the offspring and that treatment during NEC induction with I3C is protective. While the effects of AHR activation in IEC look convincing, the data do not exclude participation of other cell types, notably IEL (see comments below) and this needs to be considered in the interpretation of the data. Mechanistically the authors suggest the AHR activation upregulates miRNAs that reduce TLR4 expression.

In light of these interesting results it is disappointing that the authors – presumably in order to support their patent application for the 'ligand' A18- include in this manuscript data in Fig.6 which are not convincing, poorly integrated with the rest of the data and over-interpreted, especially in the Discussion section.

The data on A18 do not add anything to the rest of the manuscript and could be discarded without affecting the conclusions of this study. Furthermore, the pharmacokinetic parameters for A18 described in the discussion make it unsuitable for purpose, whereas I3C has been clinically validated in Phase I and II studies and is mechanistically well understood. It is not an AHR ligand, but a pro-ligand that under acidic conditions in the stomach is converted to the high affinity AHR ligand, ICZ. Interestingly, the authors cite a publication that linked acid reduction therapy to an increase in NEC when administered to premature neonates. This would make sense in the context of I3C conversion which will be inhibited in the absence of acidic conditions. A18 likewise is unlikely to be a ligand, but might be either a pro-ligand or an inhibitor of Cyp1a1 like its relative omeprazole, resulting in indirect activation of AHR by preventing metabolism of other ligands (see Wincent et al. PNAS 109, 4479 (2012)). One would hope that therapeutic trials for AHR ligand supplementation in NEC will not be held back by unwarranted emphasis on such a poor ligand candidate in favour of I3C.

Specific points:

- NEC is known to be associated with increased bacterial translocation. As the authors demonstrate considerable tissue damage in NEC-samples and increased TUNEL+ IECs, it is surprising that assessment of barrier permeability did not show any differences. Given that TLR4 downregulation in neonates is a mechanism to limit exaggerated inflammatory responses, the authors should assess whether loss of AHR in IECs during NEC leads to increased intestinal inflammation eg checking whether this corresponds with increased CD45+ cell recruitment (particularly neutrophils which are associated with NEC pathology). Fig 1h – the authors make a claim that there are TUNEL+ epithelial cells shown in this figure, but it is quite clear that the signal is also coming from the lamina propria. The figure is not of high enough resolution and should be improved. Extended Fig 1 – If this staining was meant to confirm deletion of AHR, then why does the AhRIEC image still have staining? The low quality of the IF staining and resolution of the images makes it hard to draw any conclusions regarding the efficiency of deletion.
- The authors only looked at TNF expression as marker for inflammation following I3C treatments and it might provide a stronger case for their findings if they were to include other evidence for increased inflammation (e.g. MIP-2, IL-6). This could also increase biological significance for use of AHR ligands in protection against NEC, as the histological scores shown in Fig 2k only show a modest change in NEC severity.
- The authors argue that the protective effects of I3C is primarily acting on IECs, but given the lack of statistical significance between AHR-/- and AHRIEC in Fig. 3d it is possible that other cells responsive to AHR ligands may contribute to protection against NEC. Gomez de Agüero et al (2016 Science), showed that AHR ligands are transferred through the maternal-fetal axis (also via

breastmilk until weaning) and is required for the expansion of AHR regulated immune cells such as ILC3s. IEL subsets have been noted to confer protection against NEC development and are also recruited in early life and are dependent on AHR ligands for their survival (Denning TL et al, 2017 Semin Perinatol.; Weitkamp JH et al, 2014 Plos One; Li Y et al, 2011 Cell). The authors used AHR^{LysM} and IL-22KO mice to rule out a role for the myeloid compartment and IL-22, but this does not rule out a role for IELs. They could at least check whether IELs are increased (e.g. by IF staining) upon I3C treatment.

- To strengthen the proposed mechanistic link to downregulation of TLR4 in the AHR^{-/-} enteroids (Fig. 4), the authors should check canonical downstream targets of active TLR4 signalling (e.g. phosphorylation of Nfkb signalling molecules) and confirm downregulation of TLR4 by protein quantification either through assessing surface vs intracellular levels of TLR4 by flow cytometry or by checking TLR4 quantity by WB for at least one of the key experiments.
- There are no error bars on Fig. 2k for AHR^{-/-}, AhRIEC, was a single mouse used? If so, this is not sufficient to make conclusions
- Fig.4b: how is it possible the authors obtain Cyp1a1 induction by adding I3C in vitro? I3C is not itself a ligand for AHR and requires transformation under acidic conditions in vivo. The concentration added is not mentioned – could this be contamination of I3C?
- Fig.4g: miR-146b expression does not change – overstatement in the text?
- Fig.6- see also explanations above. Why was this compound not assessed side by side with bona fide AHR ligands such as I3C derived ICZ? The latter works in pM concentrations, whereas the fact that A18 had to be added in orders of magnitude higher concentrations speaks against its quality as direct AHR ligand. Likewise for Fig.6m-q, A18 and I3C maternal administration should be compared.

Reviewer #2:

Remarks to the Author:

This is a well written manuscript from a laboratory with expertise in this field. The premise is to determine the role that maternal-fetal signaling may have on infant susceptibility to NEC. The authors have identified a potential ligand and receptor, shown that it appears relevant across species and elucidated some aspects of the potential mechanistic pathway. The statistics appear appropriate. I have a few questions, but overall found this to be a valuable and interesting manuscript.

- 1) The dose of I3C selected was 25mg/kg. How was this selected and is there a dose effect? 25mg/kg is a very large dose. I3C has been studied with regard to human breast cancer trials and the "high" human dose was closer to 6-10mg/kg.
- 2) Would also note that I3C may also have toxic effects of the embryo, in particular to the male reproductive track.
- 3) The authors studies a number of different cytokine markers of inflammation and NEC, including IL-6, iNos, and TNF alpha, but a different one was used in different figures. It would be helpful to see each of those markers across each experimental system.
- 4) The use of knock out mice and enteroids were highly compelling. Although IL-22 knock outs were used as a proxy to negate the effects of t-cells on the AHR pathway in nec, it would have been stronger if t-cell knock out mice could have been utilized.

Reviewer #3:

Remarks to the Author:

Thanks for asking me to review the manuscript by Lu et al. This important work is looking at the role of AHR and its receptors in the pathogenesis of NEC. The manuscript is well written, the figures are clearly presented, and the statistics are appropriate. I do have concerns about the lack of some baseline data and some over-reading of the data.

Specific comments:

- 1) PCR is used throughout the paper, however it is inconsistently presented. Please use a standard Y axis for PCR or at least explain in the figure legends/methods why they are different.
- 2) In figure 2I the authors use TNF expression as a marker of NEC severity. It is unclear why they chose TNF here instead of IL6 which was used in fig 1 and other instances of NEC severity.
- 3) As a general statement, there are no dose curves presented in the manuscript which makes it unclear the magnitude or dose-dependency of substances such as A18 and Ic3.
- 4) I am concerned about the specificity of the Ahr^{IEC} and Ahr^{Lys} mice. In Ext data fig 1, The Ahr^{IEC} mice which are supposed to lack Ahr signaling in the epithelia have a pretty robust fluorescent signal. Likewise, the Ahr^{Lys} mice which should only lack Ahr in myeloid cells have different epithelial staining than WT. Can the authors provide other data, or respond to this issue?
- 5) As a general statement, the authors feel that the epithelial signaling of Ahr is the predominant pathway involved with NEC. I agree that this is an important pathway and that the data presented support that. However, I am less convinced that the myeloid pathway is unimportant. Many of the data presented show no difference between ^{IEC} and ^{Lys} lines. I am OK with the authors not chasing myeloid experiments, but they need to make a serious effort in the text to not imply that the epithelial signaling is the only important pathway. Myeloid effects could be directly or indirectly impacting much of the data presented.
- 6) Similar, the A18 data was interesting, but no screens were done in myeloid cells. Isn't it possible that A18 has an equal or greater effect through that population? This is critically important for translation to clinical studies.
- 7) Breast milk is notoriously heterogeneous both between mothers and between expressions. Breast milk was used in the enteroid experiments but no mention was made if this was batched, fresh, treated, etc... Given the potential variability in breast milk content (and unclarity about the variability in breast milk), this is an important oversight.
- 8) The authors need to remove the statement in the discussion "In view of the fact that NEC almost always develops in the absence of breast milk and the presence of formula". This statement is simply incorrect. Most infants in the US now have exposure to breast milk and NEC is not just a formula problem. In addition, the citation for this statement is for a self-cited review article and not epidemiologic data to support the statement.

REVIEWER COMMENTS

Reviewer #1 (Remarks to the Author):

This is an interesting and important study evaluating the effects of AHR stimulation in pregnant mice on the development of a necrotizing enterocolitis (NEC) model with some correlative measurements of AHR expression in humans and piglets with NEC. On the whole, the effect of supplementation with the AHR pro-ligand I3C during pregnancy and the downstream effects on the newborn mice is convincingly demonstrated, suggesting that the AHR status of the mother has a strong impact on intestinal health of the offspring and that treatment during NEC induction with I3C is protective. While the effects of AHR activation in IEC look convincing, the data do not exclude participation of other cell types, notably IEL (see comments below) and this needs to be considered in the interpretation of the data. Mechanistically the authors suggest the AHR activation upregulates miRNAs that reduce TLR4 expression.

We greatly appreciate the Reviewer's kind words in describing our work as "interesting and important", and by asserting that "the downstream effects on the newborn mice is convincingly demonstrated", and that the effects of AHR activation in IEC "look convincing".

We have also now specifically addressed the potential role of IEL in detail below. In brief, we now provide evidence that IEL populations are extremely rare in the intestinal mucosa of the newborn mice with and without NEC, and their frequency is not increased after AHR activation. Most notably, we have now generated mice that lack IELs, and found that AHR activation with I3C still protects against NEC development in this IEL-deficient strain, thus excluding a role for IELs in the mechanisms of AHR protections. This data is summarized in greater detail below, and is presented in the **New Supplementary Fig. 5** and **Revised Results page 8-9**.

In light of these interesting results it is disappointing that the authors – presumably in order to support their patent application for the 'ligand' A18- include in this manuscript data in I3C6 which are not convincing, poorly integrated with the rest of the data and over-interpreted, especially in the Discussion section. The data on A18 do not add anything to the rest of the manuscript and could be discarded without affecting the conclusions of this study. Furthermore, the pharmacokinetic parameters for A18 described in the discussion make it unsuitable for purpose, whereas I3C has been clinically validated in Phase I and II studies and is mechanistically well understood. It is not an AHR ligand, but a pro-ligand that under acidic conditions in the stomach is converted to the high affinity AHR ligand, ICZ. Interestingly, the authors cite a publication that linked acid reduction therapy to an increase in NEC when administered to premature neonates. This would make sense in the context of I3C conversion which will be inhibited in the absence of acidic conditions. A18 likewise is unlikely to be a ligand, but might be either a pro-ligand or an inhibitor of Cyp1a1 like its relative omeprazole, resulting in indirect activation of AHR by preventing metabolism of other ligands (see Wincent et. Al. PNAS 109, 4479 (2012). One would hope that therapeutic trials for AHR ligand supplementation in NEC will not be held back by unwarranted emphasis on such a poor ligand candidate in favour of I3C.

Author response: Respectfully, we take a different view to the reviewer on the inclusion of the A18 data. We hold that the A18 findings in **Figure 6** provide valuable proof-of-concept data that the AHR pathway can be targeted to prevent and also treat NEC, a pathway that is hitherto unrecognized in this disease. With regards to the comments regarding our patent application, we respectfully point out that A18 is a clinically validated compound, it is a marketed drug, and it has been approved for use in patients in the US since 1995 marketed as Lansoprazole, and therefore the reviewer's negative comments about a "patent application for the ligand A18" as well as "poor PK parameters" do not apply (we do explicitly state on page 25 that "the authors have filed a patent application for the use of AHR agonists in the prevention and treatment of NEC", as would be expected for the current study, but A18 is not part of any patent filing, for the reasons stated above). Moreover, our *in vitro* data confirm that A18, and not a gastric hydrolysis product, can activate AHR (**please see Figure 6a and Figure 6c**). In aggregate, the data indicate that this compound represents a valid lead structure for further ligand optimization. We also do not suggest that A18 itself should be used in clinical trials for NEC treatment, so the concerns raised regarding effects on gastric pH really do not apply, and were highlighted by us in order to explain why further chemical modification of A18 would be required for potential clinical use in NEC trials. As for I3C, this agent, although well understood, has significant limitations as a drug for NEC i.e. very rapid half-life, very fast metabolism and excretion of I3C; levels of I3C fall below detection within 1h of oral administration¹. These factors in aggregate disqualify I3C as being a viable lead structure for drug development, and led us to identify alternate candidates that could activate AHR and prevent NEC. We have revised the text of the **discussion (page 14)** to reflect these points.

With respect to the characterization of A18's ligand properties, short of having an x-ray structure, we cannot really speculate what parts of A18 could interact with the target and in what fashion. That said, since the indole nucleus is a privileged ligand for AHR^{2,3}, it is likely that the benzimidazole core of A18 constitutes at least part of its main pharmacophore for AHR activation. This information and the appropriate references are included in the **Revised Discussion (page 14)**.

Finally, with respect to the reviewer's negative comment that the data in **Figure 6** are not "convincing", we take a different view. Specifically, we included a dose-response curve in an AHR reporter system with appropriate vehicle controls (**Figure 6b-c**). We included data that validated the ability of A18 to activate AHR *in vitro* (**Figure 6c**) and reduce TLR4 signaling and reduce NEC *in vivo* using both wild-type and AHR-deficient mice (**Figure 6d-k**). We also provided data for A18 in reducing NEC, when given either postnatally (**Figure 6h-k**) as well as *in utero* (**Figure 6p-t**). Further, we have provided data regarding the physiological relevance of A18, by showing its ability to limit TLR4 signaling in freshly harvested human ileum samples (**Figure 6l-n**). In sum, we respectfully feel that the inclusion of two models (endotoxemia and NEC), two strains of mice (wild-type and *Ahr*^{-/-} mice), as well as human ileum (which is rarely performed in validation studies such as these) constitute a set of reliable, valid and convincing data points that demonstrate the ability of the A18 chemotype to limit TLR4 signaling and prevent NEC through activation of AHR.

Specific points:

• *NEC is known to be associated with increased bacterial translocation. As the authors demonstrate considerable tissue damage in NEC-samples and increased TUNEL+ IECs, it is surprising that assessment of barrier permeability did not show any differences.*

Author response: We agree with the reviewer that NEC is associated with bacterial translocation. In the original manuscript, we measured the intestinal permeability in wild-type and *Ahr*^{-/-} mice without NEC, and determined that there were no differences in barrier permeability between these strains at baseline. In order to address the reviewer's question regarding the effects of NEC on intestinal permeability directly, we have now assessed the intestinal permeability in wild-type and *Ahr*^{-/-} mice both with and without NEC. As shown in **Supplementary Fig. 4j**, we have determined that the induction of NEC in both wild-type and *Ahr*^{-/-} mice leads to an increase in permeability as compared to control mice that do not have NEC, and there is no difference in permeability between strains. These data are included in **Revised Results pages 7-8**, and indicate that NEC induction in *Ahr*^{-/-} mice is sufficient to induce an increase in barrier permeability, but insufficient to cause the epithelial and mucosal changes that lead to NEC. It was in fact based upon these studies that excluded the potential for AHR to regulate permeability as a factor in the mechanisms of NEC protection, that we focused on the role of AHR activation in reducing TLR4 signaling or expression in the neonatal intestinal epithelium as is now clarified in **Revised Results pages 7-8**.

Given that TLR4 downregulation in neonates is a mechanism to limit exaggerated inflammatory responses, the authors should assess whether loss of AHR in IECs during NEC leads to increased intestinal inflammation eg checking whether this corresponds with increased CD45+ cell recruitment (particularly neutrophils which are associated with NEC pathology).

Author response: In direct response to the reviewer's request, we have performed the requested experiments. Specifically, we have now assessed the degree of intestinal inflammation in wild-type and *Ahr*^{-/-} mice by checking the extent of CD45 cell and neutrophils recruited to the intestinal mucosa of mice with and without NEC. As shown in **Supplemental Fig. 4k-m**, we determined that NEC significantly increased the number of CD45 cells, the number of neutrophils (Ly6G+), and the percentage of neutrophils in CD45 cells in both wild-type and *Ahr*^{-/-} mice. However, there were no significant differences in the numbers of CD45 cells and neutrophils in *Ahr*^{-/-} mice compared with wild-type mice with NEC. These data support the overall findings that the effects of AHR signaling in preventing NEC are not due to a broadscale reduction in inflammatory cells, but rather to the novel effects of AHR signaling on reducing TLR4 function in the newborn gut. These findings are included in **Supplementary Fig. 4k-m and revised Results page 8**.

Fig 1h – the authors make a claim that there are TUNEL+ epithelial cells shown in this figure, but it is quite clear that the signal is also coming from the lamina propria. The figure is not of high enough resolution and should be improved.

Authors response: We agree with the reviewer that there is also lamina propria signal. The TUNEL staining was included only to provide additional information regarding NEC severity. Given that we have reliably quantified NEC severity by displaying representative histologic

sections of the intestinal mucosa (**Fig. 1e**), as well as measurements of the blinded histological severity scores (**Fig. 1f**), and that we have calculated the expression of the proinflammatory genes *Il6* (**Fig. 1g**) and *Tnf- α* (**Fig. 1h**), we conclude that we have sufficient objective measures to quantify NEC severity, and so have elected to remove the TUNEL staining, in order to avoid confusion.

Extended Fig 1 – If this staining was meant to confirm deletion of AHR, then why does the AhRIEC image still have staining? The low quality of the IF staining and resolution of the images makes it hard to draw any conclusions regarding the efficiency of deletion.

Author response: In order to further draw conclusions regarding the efficiency of deletion of *Ahr* from the intestinal epithelium of the *Ahr^{ΔIEC}* mouse, and also from the macrophages in the *Ahr^{Δlys}* mouse, two approaches were undertaken: first, we measured the expression of *Ahr* in enteroids and in macrophages from each of the *Ahr* transgenic mouse strains, and second, we measured the functional response to I3C treatment in intestinal epithelial cells and in macrophages obtained from the *Ahr* transgenic mice, and assessed for the expression of the AHR response gene *Cyp1a1*.

Specifically, as described in **Revised Methods page 18** and **Revised Figure legends for Supplementary Fig. 1**, we first generated enteroids from the intestinal epithelium of the *Ahr^{ΔIEC}*, *Ahr^{-/-}*, *Ahr^{Δlys}* and wild-type mice. As shown in **Supplementary Fig. 1a-b**, each of these populations of enteroids express the sucrase isomaltase gene as expected but not the macrophage gene F4/80, confirming the enteroids to be of intestinal epithelial origin. Importantly, *Ahr* was not expressed in enteroids from *Ahr^{-/-}* and *Ahr^{ΔIEC}* mice, yet was robustly expressed in enteroids from *Ahr^{Δlys}* and wild-type mice, confirming the specificity of the deletion (**Supplementary Fig. 1c**). In further confirmation of these gene expression findings, we also treated enteroids from each of these strains with the AHR ligand I3C and assessed the expression of the AHR response gene *Cyp1a1*. As shown in **Supplementary Fig. 1d**, treatment of enteroids from wild-type and *Ahr^{Δlys}* mice did induce the expression of *Cyp1a1* as expected, but did not induce *Cyp1a1* in enteroids from either *Ahr^{-/-}* or *Ahr^{ΔIEC}* mice. These findings are consistent with the lack of *Ahr* expression in the enteroids from *Ahr^{-/-}* or *Ahr^{ΔIEC}* mice, and further support the specificity of the deletion strategy.

To confirm the efficiency of *Ahr* deletion in the *Ahr^{Δlys}* mice, we harvested the peritoneal macrophage-rich cells from *Ahr^{ΔIEC}*, *Ahr^{-/-}*, *Ahr^{Δlys}* and wild-type mice as described in **Revised Methods page 22**, based on the work of Layoun *et al*⁴. As shown in **Supplementary Fig. 1a-b**, and **Revised Figure Legends, Supplementary Fig. 1**, peritoneal cells from these strains did not express the intestinal marker sucrase isomaltase (**Supplementary Fig. 1a**) but did express the macrophage marker F4/80 (**Supplementary Fig. 1b**), confirming their macrophage abundance. Importantly, peritoneal cells from *Ahr^{-/-}* and *Ahr^{Δlys}* mice did not express *Ahr* (**Supplementary Fig. 1e**), while *Ahr* was expressed in peritoneal cells from wild-type and *Ahr^{ΔIEC}* mice, confirming the specificity of *Ahr* deletion in the *Ahr^{Δlys}* mice.

As an additional step to confirm the lack of *Ahr* in the *Ahr^{Δlys}* mice, we treated peritoneal cells from *Ahr^{ΔIEC}*, *Ahr^{-/-}*, *Ahr^{Δlys}* and wild-type mice with I3C and measured the expression of the AHR response gene *Cyp1a1*. As shown in **Supplementary Fig. 1f**, I3C treatment resulted in the induction of *Cyp1a1* in wild-type and *Ahr^{ΔIEC}* peritoneal cells (which express *Ahr*), but not in *Ahr^{-/-}* and *Ahr^{Δlys}* peritoneal cells (which do not).

Taken together, these functional assays confirm the cell-specific *Ahr* knockout strategy.

In view of the issues raised by the reviewer regarding the *Ahr* staining, we have removed those images in order to avoid any confusion. These data appear in the **Revised Figure Legends, Supplementary Fig 1.**

• *The authors only looked at TNF expression as marker for inflammation following I3C treatments and it might provide a stronger case for their findings if they were to include other evidence for increased inflammation (e.g. MIP-2, IL-6). This could also increase biological significance for use of AHR ligands in protection against NEC, as the histological scores shown in Fig 2k only show a modest change in NEC severity.*

Author response: We thank the Reviewer for this suggestion, and have now included other evidence of increased inflammation, including the expression of IL-6 as requested. This data now appears in **Revised Figures 1 to 6** corresponding to each NEC model.

• *The authors argue that the protective effects of I3C is primarily acting on IECs, but given the lack of statistical significance between AHR^{-/-} and AHR^{IEC} in I3C 3d it is possible that other cells responsive to AHR ligands may contribute to protection against NEC. Gomez de Agüero et al (2016 Science), showed that AHR ligands are transferred through the maternal-fetal axis (also via breastmilk until weaning) and is required for the expansion of AHR regulated immune cells such as ILC3s. IEL subsets have been noted to confer protection against NEC development and are also recruited in early life and are dependent on AHR ligands for their survival (Denning TL et al, 2017 Semin Perinatol.; Weitkamp JH et al, 2014 Plos One; Li Y et al, 2011 Cell). The authors used AHR^{LysM} and IL-22^{KO} mice to rule out a role for the myeloid compartment and IL-22, but this does not rule out a role for IELs. They could at least check whether IELs are increased (e.g. by IF staining) upon I3C treatment.*

Author response: We appreciate this thoughtful comment and suggestion. In response, we have now thoroughly assessed the possibility that IELs could play a role in the mechanisms by which AHR activation with I3C could protect against NEC development.

Specifically, we first assessed whether IELs are increased in the intestinal mucosa of the mouse pups upon I3C treatment. As shown in **New Supplementary Fig. 5a** and **Revised Results page 8-9**, approximately 90% of the IELs in the newborn mouse intestine were $\gamma\delta$ T cells, which led us to focus on this IEL subtype. As shown in **New Supplementary Fig. 5b**, there were no changes in the quantity of IELs between wild-type and *Ahr*^{-/-} pups. Moreover, administration of I3C did not increase the quantity of IELs, and there were also no significant differences on the quantity of IELs between mice without and with NEC (**New Supplementary Fig. 5c, Revised Results page 8-9**).

Next, to directly assess for a potential role for IELs in the protection by I3C for NEC, we depleted IELs by breeding ROSA-DTA mice (B6.129P2-Gt(ROSA)26Sor^{tm1(DTA)Lky/J}) with TCR δ ^{CreER} mice (B6.129S-Tcrd^{tm1.1(cre/ERT2)Zhu/J}) to generate ROSA-DTA/TCR δ ^{CreER} mice, in which the IELs can be ablated using tamoxifen. As shown in **New Supplementary Fig. 5d**, treatment with I3C did induce Cyp1a1 in the intestinal mucosa in IEL-depleted mice, confirming that AHR activation still occurred within the newborn gut in the absence of IELs. Importantly however, the administration of I3C still protected IEL-depleted pups from NEC (**New**

Supplementary Fig. 5e-h and Revised Results page 8-9), making it unlikely that the effects of I3C can be attributed to IELs.

Taken in aggregate, these findings reveal that IELs play a limited role, if any, in the mechanisms by which AHR activation with I3C protects against the development of NEC.

• *To strengthen the proposed mechanistic link to downregulation of TLR4 in the AHR^{-/-} enteroids (I3C 4), the authors should check canonical downstream targets of active TLR4 signalling (e.g. phosphorylation of Nfkb signalling molecules) and confirm downregulation of TLR4 by protein quantification either through assessing surface vs intracellular levels of TLR4 by flow cytometry or by checking TLR4 quantity by WB for at least one of the key experiments.*

Author response: We appreciate this opportunity to strengthen the mechanistic link to downregulation of TLR4 in *Ahr^{-/-}* enteroids. To do so, we have focused on NFkB signaling as suggested by the Reviewer. Specifically, we have now measured the effects of AHR activation with I3C on the translocation of NFkB from the cytoplasm to the nucleus in response to TLR4 activation by LPS in enteroids. As shown in **New Revised Fig. 5c-d**, I3C treatment significantly reduced LPS-induced NFkB translocation from the cytoplasm to the nucleus in these enteroids, providing an additional measure of TLR4 inhibition by evaluating this downstream target. This data now appears in **Revised Results page 9** and **Revised Methods page 21**.

Unfortunately, our experience with TLR4 antibodies mirrors that of the field, in that they are unreliable for SDS-PAGE and immunofluorescence.

• *There are no error bars on Fig. 2k for AHR^{-/-}, AhRIEC, was a single mouse used? If so, this is not sufficient to make conclusions*

Author response: We appreciate the Reviewer pointing this out. There were actually 7 *Ahr^{-/-}* mice and 7 *Ahr^{AIEC}* mice, all of whom developed severe NEC as is shown, and we apologize for our oversight in the lack of clarity on this point. The data now appear in **Revised Figure 2k**, in which we have replaced the original panel with a dot-plot graph.

• *I3C 4b: how is it possible the authors obtain Cyp1a1 induction by adding I3C in vitro? I3C is not itself a ligand for AHR and requires transformation under acidic conditions in vivo. The concentration added is not mentioned – could this be contamination of I3C?*

Author response: Although I3C has a relatively low affinity for AHR⁵ and often is considered as a precursor of high affinity AHR ligands, including diindolylmethane (DIM) and other oligomers, it should be noted that DIM can also spontaneously form from I3C in neutral cell culture medium *in vitro*⁶. This spontaneous *in vitro* formation of DIM may explain our consistent finding in **Figure 4b** that I3C can activate AHR *in vitro*, an observation that has certainly been made by others in other systems, i.e. I3C *in vitro* has been reported to activate AHR signaling in cultured MCF-7 human breast cancer cells as indicated by induced AHR nuclear translocation⁷, increased *CYP1A1* mRNA transcription⁸, and increased CYP1A1 protein translation⁹. Cyp1a1 induction is a well-recognized biomarker of aryl hydrocarbon receptor activation in assessing the results of large scale screening experiments using *in vitro* techniques, in support of our findings¹⁰.

The concentrations of both I3C and LPS are now included in the **Revised Figure 4b Legend**.

• *I3C4g: miR-146b expression does not change – overstatement in the text?*

Author response: The *p* value for *miR-146b* was significant at 0.030 and we apologize for forgetting to add the asterisk to this figure to denote statistical significance. This has been corrected in **Revised Figure 4g**.

I3C6- see also explanations above. Why was this compound not assessed side by side with bona fide AHR ligands such as I3C derived ICZ? The latter works in pM concentrations, whereas the fact that A18 had to be added in orders of magnitude higher concentrations speaks against its quality as direct AHR ligand. Likewise for I3C6m-q, A18 and I3C maternal administration should be compared.

Authors Response: We do not seek to make claims regarding the potential superiority of one AHR ligand over another since this is not a drug-profiling paper. Moreover, given how vastly different the chemical structures of A18 and I3C are, and the fact that A18 is a lead candidate but not intended to be used clinically in its current form, we hold that direct comparisons between A18 and I3C, while potentially interesting, do not add significantly to the paper. When A18 has been further optimized towards clinical use, we agree with the idea of performing the studies suggested by the reviewer.

Reviewer #2 (Remarks to the Author):

This is a well written manuscript from a laboratory with expertise in this field. The premise is to determine the role that maternal-fetal signaling may have on infant susceptibility to NEC. The authors have identified a potential ligand and receptor, shown that it appears relevant across species and elucidated some aspects of the potential mechanistic pathway. The statistics appear appropriate. I have a few questions, but overall found this to be a valuable and interesting manuscript.

Author response: We greatly appreciate the Reviewer's kind words in describing our work as "well written", and by mentioning that we have "expertise in this field", and in particular appreciate the comment that she/he found the work to be "valuable" and "interesting".

1) The dose of I3C selected was 25mg/kg. How was this selected and is there a dose effect? 25mg/kg is a very large dose. I3C has been studied with regard to human breast cancer trials and the "high" human dose was closer to 6-10mg/kg.

Author response: In order to address the reviewer's concern, we have now performed a dose-response study, using a range of 5mg/kg to 50mg/kg. As shown in **New Supplementary Fig. 3**, the administration of 5 or 10 mg/kg I3C did not significantly activate AHR and had little protective effect in experimental NEC, while 25 or 50 mg/kg I3C significantly activated AHR and protected mice from NEC. These findings are included in **Revised Results pages 7-8**, and the selection of an I3C dose of 25mg/kg.

With respect to the concerns regarding the dose of I3C that was used in the current study, we do agree with the Reviewer that 25 mg/kg would be a large dose for human studies, although mice show a relatively high tolerance to I3C. In fact, prior studies have shown the LD₅₀ of I3C in mice to be more than 2000 mg/kg¹¹, and 25 mg/kg or higher doses have been beneficial in several mouse models¹¹⁻¹⁴, a dose that is consistent with the current studies.

2) *Would also note that I3C may also have toxic effects of the embryo, in particular to the male reproductive track.*

Author Response: We appreciate that concern, and have now added to the **Revised Discussion page 16** the fact that I3C may have toxic effects on the male reproductive track. We have also included reference to the Wilker *et al* study which shows that exposure of pregnant rats to I3C at doses of up to 100mg/kg caused reproductive abnormalities in male offspring¹⁵ (much higher than the dose in the current study).

3) *The authors studies a number of different cytokine markers of inflammation and NEC, including IL-6, iNos, and TNF alpha, but a different one was used in different figures. It would be helpful to see each of those markers across each experimental system.*

Author Response: In response to this point, and Reviewer 1, we have now standardized each of our readouts so that we have provided the expression of *Il-6* and *Tnf-α* across each experimental system.

4) *The use of knock out mice and enteroids were highly compelling. Although Il-22 knock outs were used as a proxy to negate the effects of t-cells on the AHR pathway in NEC, it would have been stronger if t-cell knock out mice could have been utilized.*

Response: We appreciate this point. While the idea of using a T cell KO mouse to assess the specific role of T cells on the AHR pathway in NEC would indeed have been more compelling in principle, we draw attention to our paper in *JCI* that showed that T cell-deficient mice are protected from the development of NEC¹⁶, due to the critical role of Th17 cells in NEC pathogenesis. For this reason, T cell knockout mice cannot be utilized for these suggested experiments.

Reviewer #3 (Remarks to the Author):

Thanks for asking me to review the manuscript by Lu et al. This important work is looking at the role of AHR and its receptors in the pathogenesis of NEC. The manuscript is well written, the figures are clearly presented, and the statistics are appropriate. I do have concerns about the lack of some baseline data and some over-reading of the data.

Author response: we greatly appreciate the kind words of the Reviewer in describing our work as “well written”, “clearly presented”, with “appropriate statistics”.

Specific comments:

1) PCR is used throughout the paper, however it is inconsistently presented. Please use a standard Y axis for PCR or at least explain in the figure legends/methods why they are different.

Author Response: We appreciate this comment, and have now presented the PCR data in a consistent manner, using a standard, linear Y axis, throughout the paper.

2) In figure 2I the authors use TNF expression as a marker of NEC severity. It is unclear why they chose TNF here instead of IL6 which was used in fig 1 and other instances of NEC severity.

Author Response: We have now modified the Figures so as to include the expression of *Tnf- α* and *Il-6* as markers of NEC severity, along with histology, throughout the manuscript.

3) As a general statement, there are no dose curves presented in the manuscript which makes it unclear the magnitude or dose-dependency of substances such as A18 and Ic3.

Author Response: We have now included a dose-response for the protective effects of I3C on NEC severity, as described above in the response to Reviewer 2, and is described in **New Supplementary Fig. 3**, and **Revised Results pages 7-8**. Our rationale for not performing a side-by-side comparison between A18 and I3C is described above in response to Reviewer 1.

4) I am concerned about the specificity of the *Ahr*^{IEC} and *Ahr*^{Lys} mice. In Ext data fig 1, The *Ahr*^{IEC} mice which are supposed to lack *Ahr* signaling in the epithelia have a pretty robust fluorescent signal. Likewise, the *Ahr*^{Lys} mice which should only lack *Ahr* in myeloid cells have different epithelial staining than WT. Can the authors provide other data, or respond to this issue?

Author Response: We appreciate this comment and have provided additional data to address the specificity of the knockout strategy. This approach was described in detail in our response to Reviewer 1, and is now included in **New Supplementary Fig. 1** and **Revised Methods page 18**.

In brief, we harvested enteroids from the *Ahr*^{IEC} mouse and showed these do not express *Ahr* and do not induce *Cyp1a1* when treated with the AHR ligand I3C, In an important control, peritoneal macrophages from these same mice do express *Ahr* and do express *Cyp1a1*. In parallel, we harvested macrophage rich peritoneal cells from the *Ahr*^{Lys} mouse, and showed these cells do not express *Ahr* and do not induce *Cyp1a1* when treated with the AHR ligand I3C, although enteroids from these same mice do express *Ahr* and do express *Cyp1a1*. These studies confirm the specificity of deletion. We agree with the Reviewer's comment regarding the unreliable nature of the confocal immunostaining, and these panels have now been removed.

5) As a general statement, the authors feel that the epithelial signaling of *Ahr* is the predominant pathway involved with NEC. I agree that this is an important pathway and that the data presented support that. However, I am less convinced that the myeloid pathway is unimportant. Many of the data presented show no difference between ^{IEC} and ^{Lys} lines. I am OK with the authors not chasing myeloid experiments, but they need to make a serious effort in the text to not imply that the epithelial signaling is the only important pathway. Myeloid effects could be

directly or indirectly impacting much of the data presented.

Author response: We agree with the Reviewer's point that we cannot fully exclude a role for myeloid cells. That said, we have made a serious effort throughout the text, in multiple places, to indicate that myeloid cells could play a role. See for example the **Revised Discussion page 14**, and **page 15**.

6) Similar, the A18 data was interesting, but no screens were done in myeloid cells. Isn't it possible that A18 has an equal or greater effect through that population? This is critically important for translation to clinical studies.

Author Response: Once we have modified A18 to a point at which it could be used clinically, additional screens in myeloid cells will certainly be important. This point is captured in the **Revised Discussion, page 16**.

7) Breast milk is notoriously heterogeneous both between mothers and between expressions. Breast milk was used in the enteroid experiments but no mention was made if this was batched, fresh, treated, etc... Given the potential variability in breast milk content (and unclarity about the variability in breast milk), this is an important oversight.

Author Response: We apologize for our oversight, and now state in the **Revised Methods page 19** that the breast milk was obtained from a single donor.

8) The authors need to remove the statement in the discussion "In view of the fact that NEC almost always develops in the absence of breast milk and the presence of formula". This statement is simply incorrect. Most infants in the US now have exposure to breast milk and NEC is not just a formula problem. In addition, the citation for this statement is for a self-cited review article and not epidemiologic data to support the statement.

Author Response: We have removed the original statement (with apologies), and have now cited epidemiologic data in support of the fact that "absence of breast milk is a major risk factor for NEC"¹⁷⁻²⁰. This statement appears in the **Revised Discussion page 14**.

It is our sincere hope that having addressed each of the concerns raised by the Reviewers and the editorial board, the work is now suitable for publication in *Nature Communications*.

Sincerely,

David Hackam, md, PhD
Professor of surgery, cell biology and pediatrics
Johns Hopkins University.

References

1. Anderton, M. J. *et al.* Pharmacokinetics and tissue disposition of indole-3-carbinol and its acid condensation products after oral administration to mice. *Clin. Cancer Res.* (2004).

- doi:10.1158/1078-0432.CCR-04-0163
2. Bessede, A. *et al.* Aryl hydrocarbon receptor control of a disease tolerance defence pathway. *Nature* **511**, 184–190 (2014).
 3. Denison, M. S. & Faber, S. C. And now for something completely different: Diversity in ligand-dependent activation of Ah receptor responses. *Current Opinion in Toxicology* (2017). doi:10.1016/j.cotox.2017.01.006
 4. Layoun, A., Samba, M. & Santos, M. M. Isolation of murine peritoneal macrophages to carry out gene expression analysis upon toll-like receptors stimulation. *J. Vis. Exp.* (2015). doi:10.3791/52749
 5. Jellinck, P. H. *et al.* Ah receptor binding properties of indole carbinols and induction of hepatic estradiol hydroxylation. *Biochem. Pharmacol.* (1993). doi:10.1016/0006-2952(93)90258-X
 6. Bradlow, H. L. & Zeligs, M. A. Diindolylmethane (DIM) spontaneously forms from indole-3-carbinol (I3C) during cell culture experiments. *In Vivo (Brooklyn)*. (2010).
 7. Marconett, C. N. *et al.* Indole-3-carbinol triggers aryl hydrocarbon receptor-dependent estrogen receptor (ER) α protein degradation in breast cancer cells disrupting an ER α GATA3 transcriptional cross-regulatory loop. *Mol. Biol. Cell* (2010). doi:10.1091/mbc.E09-08-0689
 8. Ociepa-Zawal, M., Rubiś, B., Łaciński, M. & Trzeciak, W. H. The effect of indole-3-carbinol on the expression of CYP1A1, CYP1B1 and AhR genes and proliferation of MCF-7 cells. *Acta Biochim. Pol.* (2007). doi:10.18388/abp.2007_3276
 9. Tiwari, R. K., Guo, L., Bradlow, H. L., Telang, N. T. & Osborne, M. P. Selective responsiveness of human breast cancer cells to indole-3-carbinol, a chemopreventive agent. *J. Natl. Cancer Inst.* (1994). doi:10.1093/jnci/86.2.126
 10. Hu, W., Sorrentino, C., Denison, M. S., Kolaja, K. & Fielden, M. R. Induction of Cyp1a1 is a nonspecific biomarker of aryl hydrocarbon receptor activation: Results of large scale screening of pharmaceuticals and toxicants in vivo and in vitro. *Mol. Pharmacol.* (2007). doi:10.1124/mol.106.032748
 11. Hajra, S., Patra, A. R., Basu, A. & Bhattacharya, S. Prevention of doxorubicin (DOX)-induced genotoxicity and cardiotoxicity: Effect of plant derived small molecule indole-3-carbinol (I3C) on oxidative stress and inflammation. *Biomed. Pharmacother.* (2018). doi:10.1016/j.biopha.2018.02.088
 12. Machijima, Y. *et al.* Anti-adult T-cell leukemia/lymphoma effects of indole-3-carbinol. *Retrovirology* (2009). doi:10.1186/1742-4690-6-7
 13. Lubet, R. A. *et al.* Effects of 5,6-benzoflavone, indole-3-carbinol (I3C) and diindolylmethane (DIM) on chemically-induced mammary carcinogenesis: Is DIM a substitute for I3C? *Oncol. Rep.* (2011). doi:10.3892/or.2011.1316
 14. Singh, N. P. *et al.* Dietary Indoles Suppress Delayed-Type Hypersensitivity by Inducing a Switch from Proinflammatory Th17 Cells to Anti-Inflammatory Regulatory T Cells through Regulation of MicroRNA. *J. Immunol.* (2016). doi:10.4049/jimmunol.1501727
 15. Wilker, C., Johnson, L. & Safe, S. Effects of developmental exposure to indole-3-carbinol or 2,3,7,8-tetrachlorodibenzo-p-dioxin on reproductive potential of male rat offspring. *Toxicol. Appl. Pharmacol.* (1996). doi:10.1006/taap.1996.0261
 16. Egan, C. E. *et al.* Toll-like receptor 4-mediated lymphocyte influx induces neonatal necrotizing enterocolitis. *J Clin Invest* **126**, 495–508 (2016).
 17. Herrmann, K. & Carroll, K. An Exclusively Human Milk Diet Reduces Necrotizing

- Enterocolitis. *Breastfeed. Med.* (2014). doi:10.1089/bfm.2013.0121
18. Kantorowska, A. *et al.* Impact of donor milk availability on breast milk use and necrotizing enterocolitis rates. *Pediatrics* (2016). doi:10.1542/peds.2015-3123
 19. Altobelli, E., Angeletti, P. M., Verrotti, A. & Petrocelli, R. The impact of human milk on necrotizing enterocolitis: A systematic review and meta-analysis. *Nutrients* (2020). doi:10.3390/nu12051322
 20. Nolan, L. S., Parks, O. B. & Good, M. A review of the immunomodulating components of maternal breast milk and protection against necrotizing enterocolitis. *Nutrients* (2020). doi:10.3390/nu12010014

Reviewers' Comments:

Reviewer #1:

Remarks to the Author:

The authors have provided a comprehensive response to my comments. The revised manuscript contains a substantial amount of added new data that address the major concerns and in my view have adequately dealt with all points that were raised.

Reviewer #2:

Remarks to the Author:

Thank you for your thoughtful responses. I have no additional concerns.

Reviewer #3:

Remarks to the Author:

Thanks for addressing my comments. I have no further suggestions/critiques.

REVIEWERS' COMMENTS

Reviewer #1 (Remarks to the Author):

The authors have provided a comprehensive response to my comments. The revised manuscript contains a substantial amount of added new data that address the major concerns and in my view have adequately dealt with all points that were raised.

Author – we appreciate these kind words.

Reviewer #2 (Remarks to the Author):

Thank you for your thoughtful responses. I have no additional concerns.

Author – we appreciate these kind words.

Reviewer #3 (Remarks to the Author):

Thanks for addressing my comments. I have no further suggestions/critiques.

Author – we appreciate these kind words.